# Q-Tuning: Continual Queue-based Prompt Tuning for Language Models

## ABSTRACT

This paper introduces **Q-tuning**, a novel approach for continual prompt tuning that enables the lifelong learning of a pretrained language model on a sequence of tasks. For each new task, Q-tuning trains a task-specific prompt by adding it to the prompt queue consisting of the prompts from older tasks. To better transfer the knowledge of older tasks, we design an ensemble mechanism that reweighs previous prompts in the queue with a learnable low-rank matrix that reflects their relevance to the current task. To facilitate training and inference with manageable complexity, once the prompt queue reaches its maximum capacity, we leverage a PCA-based eviction rule to reduce the queue's size, allowing the newly trained prompt to be added while preserving the primary knowledge of older tasks. In order to mitigate the accumulation of information loss caused by the eviction, we additionally propose a globally shared prefix prompt and a memory retention regularization based on the information theory. Extensive experiments demonstrate that our approach outperforms the state-of-the-art methods substantially on both short and long task sequences. Moreover, our approach enables lifelong learning on an extremely long task sequence while requiring only $\mathcal{O}(1)$ complexity for training and inference, which could not be achieved by existing technologies.

## 1 INTRODUCTION

In recent years, pretrained language models (LMs) have achieved huge success in natural language processing (Brown et al., 2020; Thoppilan et al., 2022; OpenAI, 2023), which popularizes the pretraining-finetuning pipeline in applications. However, with the ever-growing parameter scale of modern LMs (*e.g.*, GPT-4 that may have 1.76 trillion parameters (Wiki, 2023)), it becomes increasingly difficult to finetune the whole model, leading to the extensive attention to parameter-efficient finetuning (PEFT) technologies. *Prompt tuning* (PT) (Liu et al., 2022) has recently emerged as a leading PEFT solution. PT trains soft prompts and prepends them to the input of LMs, while keeping the LM parameters frozen. Existing works (Lester et al., 2021; Liu et al., 2023) have shown that PT can achieve performance on par with finetuning, while requiring less than 0.01% of the total trainable parameters. The effectiveness of PT has inspired its use in adapting pretrained LMs to different applications. Notably, PT can be used as a key methodology for learning new tasks that typically arrive in a *sequential* fashion, which extends PT to the continual learning (CL) paradigm and leads to the so-called continual prompt tuning (CPT). Such CL capability can benefit many real-world applications that require lifelong learning.

However, as a subfield of CL, CPT encounters technical challenges akin to those faced by traditional CL methods, including the well-known *catastrophic forgetting* (CF) and *forward knowledge transfer* (FKT). CF mitigation aims to enable a model to learn and adapt to new information overtime without forgetting previous knowledge. Approaches such as regularization based methods (Zenke et al., 2017; Schwarz et al., 2018) and memory-replay based methods (Bang et al., 2021; Lin et al., 2022) have been proposed to solve the CF problem. Unlike these traditional CL methods, CPT lends itself readily to address the CF issue (Zhu et al., 2022; Razdaibiedina et al., 2023) by cheaply saving the prompts for each task and reusing them for their corresponding tasks during inference. Nevertheless, how to empower FKT in CPT remains under-explored.

In an attempt to overcome the challenges in CPT, Razdaibiedina et al. (2023) proposed ProgPrompt, which progressively adds the newly trained prompt to a prompt list that maintains all previously

trained prompts. ProgPrompt achieves FKT by appending previous prompts as inputs during the learning of a new task. However, a key limitation of ProgPrompt is the infinitely increasing prompt list. Given $N$ tasks, this prompt list grows linearly at a rate of $\mathcal{O}(N)$ and leads to an $\mathcal{O}(N^2)$ complexity for transformer (Vaswani et al., 2017) based models. Therefore, the training and inference cost will become intractable as $N$ increases and exceeds a finite computation resource limit.

In this paper, we overcome the aforementioned challenge by proposing a new continual prompt tuning technology named *Queue-based prompt tuning* (**Q-tuning**). Q-tuning manages a *Queue-based prompt* (**Q-prompt**), which is stored in a *finite-size* data buffer. During the learning of a new task, Q-tuning trains a new prompt combined with a fixed Q-prompt that stores all previously learned prompts. Upon the completion of tuning for a new task, the latest trained prompt will be added to the Q-prompt for the tuning of the next task. Once the number of tasks exceeds the queue-size limit, we will remove less informative prompts according to a principal component analysis (PCA) based dequeue rule. This endows Q-tuning with the ability to perform lifelong prompt tuning on extremely long task sequences. Our key contributions and results can be summarized as follows:

- We propose a continual prompt tuning method called Q-tuning that, to our knowledge, is the first technique for achieving lifelong learning on extremely long task sequences through prompt tuning. Our Q-tuning maintains a prompt queue coupled with a dynamic low-rank queue ensemble matrix, where the ensemble matrix is optimized to capture the importance of the enqueued prompts. This queue ensemble strategy induces a new prompt tuning strategy to enhance FKT.

- Once the number of tasks exceeds the size limit of Q-prompt, we apply a novel dequeue rule based on PCA to extract and retain the most informative prompts in Q-prompt for subsequent prompt tuning. In addition, to mitigate the impact of information loss due to dequeuing, we devise a global shared prefix prompt with a memory retention (MR) technique that can be continuously updated by each incoming task to compensate for the information loss in the trimmed prompt queue.

- We conduct extensive experiments to demonstrate the successful applications of our proposed Q-tuning on both short and long sequence benchmark tasks. Q-tuning outperforms all the competing CL methods by a large margin. In addition, Q-tuning highlights its ability to facilitate lifelong learning. For instance, our experiments on extremely long learning sequences consisting of 70 disjoint tasks have shown a 30% accuracy improvement over the standard prompt tuning method.

## 2 RELATED WORK

**1) Continual Learning:** Continual Learning (CL), also known as lifelong learning, is to learn from a stream of different tasks arriving sequentially. The goal of CL is to prevent the CF problem (Kemker et al., 2018) and achieve knowledge transfer (Ke et al., 2021). Existing CL approaches can be divided into three categories: 1) Memory-based methods (Shin et al., 2017; Bang et al., 2021; Lin et al., 2022; Ermis et al., 2022) that store previous data and replay them when training on the next task to mitigate CF issue; 2) Regularization-based methods (Kirkpatrick et al., 2017; Zenke et al., 2017; Schwarz et al., 2018) that apply an additional regularization loss to constrain the update of parameters which are less important to learning new tasks; 3) Architecture-based methods that dynamically expand the network capacity (Rusu et al., 2016; Yoon et al., 2018) or train task-specific parameters (Yoon et al., 2020) on new tasks and fix parameters for old tasks to prevent forgetting. However, these methods, which require finetuning all model parameters, are too expensive to put into practice for large-scale models with an astronomical number of parameters, such as large language models (LLMs).

**2) Prompt Tuning:** Prompt tuning (Lester et al., 2021; Karimi Mahabadi et al., 2021; Li & Liang, 2021; Gu et al., 2022; Jia et al., 2022; Wang et al., 2023a; 2022; Smith et al., 2023; Yin et al., 2022) is a lightweight approach to finetune an LLM model for a target task, which only requires optimizing a series of virtual tokens (a.k.a "soft prompt") instead of updating the entire model. It has been shown that, by only training a small subset of parameters, prompt tuning can achieve the same or even better performance than training a full model, especially when requiring adaptation to a new task with limited data. In prompt tuning, a trainable soft prompt $\theta_{\mathcal{P}}$ is prepended to the input text $\mathbf{x}$ while keeping other parameters frozen. In this case, the combined model parameters include trainable prompt parameters $\theta_{\mathcal{P}}$ and parameters $\theta_{\mathcal{M}}$ of a fixed pretrained model $\mathcal{M}$. Given the task $\mathcal{T} = (\mathcal{X}, \mathcal{Y})$ consisting of training pairs $(\mathbf{x}, \mathbf{y})$, the objective of prompt tuning can be written as:

$$\max_{\theta_{\mathcal{P}}} \sum_{(\mathbf{x}, \mathbf{y}) \in \mathcal{T}} \log p(\mathbf{y}|\mathbf{x}; \theta_{\mathcal{M}}, \theta_{\mathcal{P}}). \tag{1}$$

**3) Continual Prompt Tuning:** Prompt tuning has recently been adapted to the continual learning domain (Qin & Joty, 2021; Zhu et al., 2022; Liang et al., 2023; Wang et al., 2023b; Razdaibiedina et al., 2023; Khan et al., 2023). To enable knowledge transfer, CPT combines the advantages of both prompt tuning and CL. ProgPrompt, the current state-of-the-art method of CPT proposed by Razdaibiedina et al. (2023), maintains a progressively increasing prompt list that sequentially concatenates new soft prompts with previously learned prompts. Given the continually increased task set $\mathcal{T} = \{(\mathcal{X}^1, \mathcal{Y}^1), (\mathcal{X}^2, \mathcal{Y}^2), \dots, (\mathcal{X}^i, \mathcal{Y}^i)\}$, where $\mathcal{T}^i = (\mathcal{X}^i, \mathcal{Y}^i)$ denotes the training pairs on $i$-th task, ProgPrompt aims to progressively train an increased prompt list $[\theta_{\mathcal{P}}^1, \theta_{\mathcal{P}}^2, \dots, \theta_{\mathcal{P}}^i]$, where $[\cdot, \cdot]$ denotes the concatenation operation. For each task, only the newly appended prompt is trainable, while the previously trained prompts are fixed. The objective for the $i$-th task can be written as:

$$\max_{\theta_{\mathcal{P}}^i} \sum_{(\mathbf{x}^i, \mathbf{y}^i) \in \mathcal{T}^i} \log p(\mathbf{y}^i | \mathbf{x}^i; \theta_{\mathcal{M}}, \underbrace{[\theta_{\mathcal{P}}^1, \theta_{\mathcal{P}}^2, \dots, \theta_{\mathcal{P}}^i]}_{\text{increasing prompt list}}). \tag{2}$$

This method can achieve FKT without data replay by keeping previous prompts as input for learning a new task. However, this solution has a key limitation that prevents its sustainable adoption in practice. Suppose that the total number of continually learned tasks is $N$. The training and inference complexity of maintaining the prompt list scales as $\mathcal{O}(N^2)$ for transformer based models. When $N$ grows asymptotically (*i.e.*, the model is set as a lifelong learner), training the extremely long prompt list becomes intractable due to the finite system resources. Moreover, since both the cached prompts in the list and the pretrained models remain frozen when learning a new task, the contribution of each fixed prompt to learning the new task lacks adaptive adjustment. Inspired by the memory management (Davis & Zhong., 2017) system of the human brain, we introduce Q-tuning, which solves the aforementioned quadratic complexity problem by dynamically updating the prompt queue to maintain the learned knowledge and a queue ensemble strategy to enhance knowledge transfer.

## 3 THE Q-TUNING APPROACH

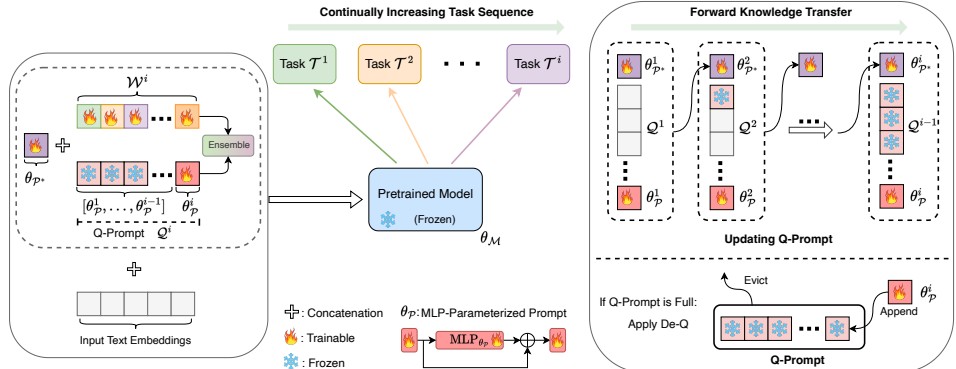

Figure 1: The overall framework of the proposed **Q-tuning** technology. Given a continually growing-up task sequence, we propose a prompt queue (Q-prompt) and a globally *shared* prefix prompt $\theta_{\mathcal{P}*}^i$ to achieve the forward knowledge transfer, where the superscript of $\theta_{\mathcal{P}*}^i$ denotes the $i$-th status. Moreover, we adopt a queue ensemble method to dynamically adjust the contribution of each fixed prompt $[\theta_{\mathcal{P}}^1, \theta_{\mathcal{P}}^2, \dots, \theta_{\mathcal{P}}^{i-1}]$ in Q-prompt by using a rank-one matrix $\mathcal{W}^i$. We parameterize the trainable soft prompt by a two-layer residual MLP. If the length of the Q-prompt exceeds the limit, we apply a De-Q rule to discard less informative prompts in the queue.

### 3.1 Q-PROMPT AND UPDATE RULE

**Q-prompt:** Fig. 1 illustrates the overall framework of the proposed Q-tuning technique. In Q-tuning, we add a new trainable prompt to a prompt queue $\mathcal{Q}$ that stores all previously trained prompts for old tasks. This updated $\mathcal{Q}$ associated with a globally shared prompt will be tuned for the new task, while keeping the prior prompts in $\mathcal{Q}$ frozen. This progressively appending approach enables forward knowledge transfer as the old task's information is saved in the Q-prompt. We let $C = l \times \mathcal{Q}_{\text{size}}$ denote the maximum capacity of the Q-prompt $\mathcal{Q}$, where $l$ is the length of a single prompt per task and $\mathcal{Q}_{\text{size}}$ is the maximum number of prompts in the queue. When reaching the capacity limit of $\mathcal{Q}$, the prompt queue will be trimmed using an eviction rule to remove less informative prompts and append new trainable prompts for future tasks.

**Q-prompt Ensemble:** In Q-tuning, all prompts in the memory (*i.e.*, the prompt queue $\mathcal{Q}$), as well as the pretrained LM model, are frozen when learning a new task. Consequently, the LM model will be forced to take these fixed prompts in the queue as inputs without incorporating their relevance to the current task, leading to sub-optimal performance. To address this problem, we propose a dynamic prompt ensemble mechanism. For task $i$, we use a trainable matrix $\mathcal{W}^i \in \mathbb{R}^{c^i \times d}$, which is of the same dimension as the Q-prompt $\mathcal{Q}^i$, to scale $\mathcal{Q}^i$ by $\mathcal{W}^i \circ \mathcal{Q}^i$ ($\circ$ denotes the Hadamard product). Here, for task $i$, we denote the total prompt length of $\mathcal{Q}^i$ by $c^i = l \times i$. Since directly optimizing a large-scale matrix of size $c^i \times d$ is costly, we propose a low-rank multiplicative method inspired by Aghajanyan et al. (2021); Wang et al. (2023a). The weight matrix $\mathcal{W}^i$ can be expressed as $\mathcal{W}^i = \mathbf{u}_i \otimes \mathbf{v}_i^{\mathrm{T}}$, where $\mathbf{u}_i \in \mathbb{R}^{c^i}$, $\mathbf{v}_i \in \mathbb{R}^d$ and $\otimes$ denotes the outer product. Clearly, $\mathcal{W}^i$ is a rank-one matrix and the number of trainable parameters is reduced to $c^i + d \ll c^i \times d$. We jointly optimize the newly appended prompt $\theta_{\mathcal{P}}^i$ and the low-rank ensemble matrix $\mathcal{W}^i$ by maximizing the cross-entropy loss as follows:

$$\max_{\theta_{\mathcal{P}}^i, \mathcal{W}^i} \sum_{(\mathbf{x}^i, \mathbf{y}^i) \in \mathcal{T}^i} \log p(\mathbf{y}^i | \mathbf{x}^i; \theta_{\mathcal{M}}, \mathcal{W}^i \circ \underbrace{\mathcal{Q}^i(\theta_{\mathcal{P}}^1, \cdots, \theta_{\mathcal{P}}^i)}_{\text{maximum length is } l \times \mathcal{Q}_{\text{size}}} )), \tag{3}$$

where only the new added prompt $\theta_{\mathcal{P}}^i$ and the weight matrix $\mathcal{W}^i$ for the $i$-th task are trainable.

**De-Q Rule:** Our Q-prompt design allows appending newly trained prompts until reaching the maximum length. Once the Q-prompt is full (denoted by $\mathcal{Q}_C$), a dequeuing (De-Q) rule is executed to reduce the length of $\mathcal{Q}_C$ to $C - l$ so as to add the new prompt for the new task. However, this leads to a key question: *how to retain the most useful prompt information after trimming the Q-prompt?* Straightforward De-Q rules include random eviction and first in first out (FIFO). However, these simple rules may discard valuable information in the queue, resulting in negative impacts on FKT.

An alternative solution is to measure the correlation between a new task and the old tasks, similar to Zhu et al. (2022), and remove the most task-irrelevant prompts from the queue to learn the new task. However, this approach requires extra computing resources to maintain the data buffer of old tasks and the quantitative correlation of different tasks is hard to define. To address this problem, we introduce a simple yet effective De-Q rule named DQ-PCA based on principal component analysis (PCA) (Shlens, 2014). Specifically, we first calculate the centered Q-prompt $\tilde{\mathcal{Q}}_C \in \mathbb{R}^{C \times d}$ with a zero mean: $\tilde{\mathcal{Q}}_C = \mathcal{Q}_C - \text{mean}(\mathcal{Q}_C)$. Then we perform singular value decomposition (SVD). We extract the first $C - l$ principal components to obtain the trimmed Q-prompt $\tilde{\mathcal{Q}}_{C-l} \in \mathbb{R}^{(C-l) \times d}$ and enqueue the new trainable $\theta_{\mathcal{P}}^i \in \mathbb{R}^{l \times d}$. This process can be written as follows:

$$\text{SVD}(\tilde{\mathcal{Q}}_C) = U \Sigma V^{\mathrm{T}}, \ \tilde{\mathcal{Q}}_{C-l} = \Sigma_{C-l} V_{C-l}^{\mathrm{T}}, \ \mathcal{Q}_C \xleftarrow{\text{Update}} \tilde{\mathcal{Q}}_{C-l} \oplus \theta_{\mathcal{P}}^i, \tag{4}$$

where $\oplus$ denotes the concatenation operation $[\tilde{\mathcal{Q}}_{C-l}, \theta_{\mathcal{P}}^i]$, $U \in \mathbb{R}^{C \times C}$ is the matrix consisting of the left singular vectors, $\Sigma \in \mathbb{R}^{C \times d}$ is the diagonal matrix formed by the singular values in decreasing order and $V^{\mathrm{T}}$ is the matrix of right singular vectors. The matrix $V_{C-l}^{\mathrm{T}}$ is formed by the top $C - l$ principle row vectors of $V^{\mathrm{T}}$ and $\Sigma_{C-l} \in \mathbb{R}^{(C-l) \times (C-l)}$ denotes the diagonal matrix with the top $C - l$ singular values. When the length of the Q-prompt exceeds $C$, it will trigger the DQ-PCA to shrink the Q-prompt's length to $C - l$. As a result, Q-tuning achieves an $\mathcal{O}(1)$ training and inference complexity instead of $\mathcal{O}(N^2)$ for transformer-based LMs, thereby enabling low-cost lifelong learning[1].

### 3.2 Prefix Prompt for Global Knowledge Sharing

Although DQ-PCA is able to minimize the information loss due to the eviction in Q-prompt by keeping the most useful information of previous prompts, information loss will be inevitably accumulated as the number of tasks grows larger. To avoid such loss, we introduce a globally shared prefix prompt $\theta_{\mathcal{P}*}$. This prefix prompt is appended to the head of the Q-prompt and continually trained across all the tasks, so that it can aggregate the global information. However, naively training the shared prompt $\theta_{\mathcal{P}*}$ continuously across the tasks will lead to dominance by the newest task, hence causing the forgetting of the old knowledge. To address this limitation, we propose a *memory retention* (MR)

---

[1]For example, on a single NVIDIA V100 GPU (32GB) with the same training setting as ProgPrompt (Razdaibiedina et al., 2023), Q-tuning can easily handle an extremely long 70-task sequence, while ProgPrompt fails due to memory overflow (cf. our experiments).

regularization by maximizing the overlapping information between the shared prefix prompt and the learned knowledge from old tasks. For each task $i$, we formulate the maximization problem as:

$$\max_{\theta^i_{\mathcal{P}*}} I(\underbrace{p(\mathbf{y}^i|\mathbf{x}^i; \theta_{\mathcal{M}}, \theta^i_{\mathcal{P}*})}_{p(\xi^i)}; \underbrace{p(\mathbf{y}^i|\mathbf{x}^i; \theta_{\mathcal{M}}, \mathcal{W}^{i-1} \circ [\theta^{i-1}_{\mathcal{P}*}, \mathcal{Q}^{i-1}])}_{p(\xi^{i-1})}), \quad (5)$$

where $I(\cdot, \cdot)$ represents the mutual information between two random variables, $\theta^i_{\mathcal{P}*}$ denotes the shared prompt to be learnt for $i$-th task, $\theta^{i-1}_{\mathcal{P}*}$ is the shared prompt learnt until task $i-1$, and $\mathcal{Q}^{i-1}$ denotes the Q-prompt until task $i-1$. The second term $p(\xi^{i-1})$ in Eq. (5) represents the old knowledge learnt before the $i$-th task, provided by the shared $\theta^{i-1}_{\mathcal{P}*}$ and the Q-prompt $\mathcal{Q}^{i-1}$. Maximizing Eq. (5) can transfer the knowledge modeled by $p(\xi^{i-1})$ to current shared prompt $\theta^i_{\mathcal{P}*}$. The benefit of this knowledge transfer is that, if the Q-prompt $\mathcal{Q}^{i-1}$ at task $i-1$ reaches its maximum length $C$, $\theta^i_{\mathcal{P}*}$ can compensate the information loss caused by trimming $\mathcal{Q}^{i-1}$. As a result, when we continue to move from task $i$ to $i+1$, although the information of $\mathcal{Q}^i$ is no longer complete due to the shrinkage of $\mathcal{Q}^{i-1}$, the full information prior to task $i+1$ can be represented by the union of $\mathcal{Q}^i$ and $\theta^i_{\mathcal{P}*}$.

To solve the mutual information $I(p(\xi^i); p(\xi^{i-1}))$ in Eq. (5), we adopt the mutual information estimator[2] (Hjelm et al., 2018; Poole et al., 2019) based on the Jensen-Shannon divergence (JSD), which satisfies

$$I(p(\xi^i); p(\xi^{i-1})) := \mathcal{D}_{\text{JSD}}(\mathbf{J}; \mathbf{M}) \geq \mathbb{E}_{z \sim \mathbf{J}}\left[-\sigma(-\mathcal{F}_\omega(z))\right] - \mathbb{E}_{z' \sim \mathbf{M}}\left[\sigma(\mathcal{F}_\omega(z'))\right], \quad (6)$$

where the $\mathbf{J} = p(\xi^i, \xi^{i-1})$ and $\mathbf{M} = p(\xi^i)p(\xi^{i-1})$ are the joint and the product of marginals of the random variables $\xi^i$ and $\xi^{i-1}$, respectively, and $\sigma(t) = \log(1 + e^t)$. $\mathcal{F}_\omega$ is a discriminator function (Nowozin et al., 2016) modeled by an auxiliary neural network with parameters $\omega$.

### 3.3 OBJECTIVE FUNCTION OF Q-TUNING

Given the $i$-th classification task, the training objective of Q-tuning is defined as:

$$\mathcal{L}_{\mathcal{Q}}(\theta^i_{\mathcal{P}*}, \theta^i_{\mathcal{P}}, \mathcal{W}^i) = -\sum_{(\mathbf{x}^i, \mathbf{y}^i) \in \mathcal{T}^i} \log p(\mathbf{y}^i|\mathbf{x}^i; \theta_{\mathcal{M}}, \theta^i_{\mathcal{P}*}, \mathcal{W}^i \circ \mathcal{Q}^i(\theta^1_{\mathcal{P}}, \cdots, \theta^i_{\mathcal{P}})), \quad (7)$$

where $\mathcal{T}^i$ denotes the data streams of the $i$-th task. The pretrained model $\theta_{\mathcal{M}}$ and all the enqueued prompts prior to $i$-th task are fixed. The trainable parameters include the shared prefix prompt $\theta^i_{\mathcal{P}*}$, the newly appended prompt $\theta^i_{\mathcal{P}}$ and the queue ensemble matrix $\mathcal{W}^i$.

For the prefix prompt $\theta^i_{\mathcal{P}*}$, we enable its capability for memorizing the knowledge of old tasks with the MR regularization defined by Eq. (5). According to Eq. (6), we can maximize the lower bound of the mutual information, which can be rewritten as minimizing a loss $\mathcal{L}_{\text{MR}}$ with respect to $\theta^i_{\mathcal{P}*}$:

$$\mathcal{L}_{\text{MR}}(\theta^i_{\mathcal{P}*}) = -\mathbb{E}_{z \sim \mathbf{J}}\left[-\sigma(-\mathcal{F}_\omega(z))\right] + \mathbb{E}_{z' \sim \mathbf{M}}\left[\sigma(\mathcal{F}_\omega(z'))\right], \quad (8)$$

where $\mathbf{J}$ and $\mathbf{M}$ are defined in Eq. (5) and Eq. (6). The MLP-based discriminator $\mathcal{F}_\omega(\cdot)$ consists of two 512-unit hidden layers. To optimize Eq. (8) on a given finite training data set, we approximate the expectations using minibatch samples as in Belghazi et al. (2018).

Putting all things together, we obtain the overall loss:

$$\mathcal{L}_{total} = \mathcal{L}_{\mathcal{Q}}(\theta^i_{\mathcal{P}*}, \theta^i_{\mathcal{P}}, \mathcal{W}^i) + \eta \mathcal{L}_{\text{MR}}(\theta^i_{\mathcal{P}*}), \quad (9)$$

where $\eta$ is called "memory factor" which is used to weigh the contribution of $\mathcal{L}_{\text{MR}}$. When the number of tasks $N \leq C$, we set $\eta = 0$, whereas if $N > C$, we set $\eta > 0$. We empirically find the best $\eta$ as reported in Table 12 of Appendix D. Algorithm 1 summarizes the Q-tuning algorithm.

## 4 EXPERIMENT SETTINGS

### 4.1 DATASETS AND BASELINE METHODS

**Datasets:** Following Razdaibiedina et al. (2023), we evaluate the proposed Q-tuning on a short-sequence benchmark and a long-sequence benchmark. In the short-sequence CL benchmark, we

---

[2]More details about the deviation of the mutual information estimator can be found in Appendix B.

adopt five text classification datasets by Zhang et al. (2015), including YP reviews, Amazon reviews, DBpedia, Yahoo Answers, and AG News. To validate our method's efficacy on different model backbones, we adopt the T5-large model (an encoder-decoder model) and the BERT-base model (an encoder-only model) for evaluation. To demonstrate that the Q-tuning is robust against the order of received tasks, for the experiments with T5, we use three different orders (*i.e.*, Orders 1∼3[3]) composed of the AG News, Amazon, Yahoo and DBpedia datasets by following the few-shot CL setting as in Qin & Joty (2021); Razdaibiedina et al. (2023). For the BERT-based experiments, we use four different orders (*i.e.*, Orders 4∼7[3]) including all the above five tasks, and we use the same train and test split as IDBR (Huang et al., 2021) including 115,000 training and 7,600 test examples.

In addition, to evaluate our model on a more realistic CL scenario with a long sequence of tasks, following Razdaibiedina et al. (2023), we choose a long-sequence CL benchmark setting with 15 tasks, which consists of the aforementioned five datasets from the short-sequence CL benchmark, four tasks from GLUE benchmark (MNLI, QQP, RTE, SST2) by Wang et al. (2018), five tasks from SuperGLUE benchmark by Wang et al. (2019) (WiC, CB, COPA, MultiRC, BoolQ), and IMDB movie reviews dataset (Maas et al., 2011). We use three different orders (*i.e.*, Orders 8∼10[3]). Lastly, to mimic the lifelong learning scenario, we further add the Banking77 dataset (Casanueva et al., 2020), the Emotion dataset (Saravia et al., 2018), the rest datasets (WNLI, COLA and QNLI ) from the GLUE benchmark, and WSC from the SuperGLUE benchmark. We construct a benchmark with a long sequence of 70 tasks by splitting the datasets with over 4 classes into *disjoint* subsets[4]. Following Razdaibiedina et al. (2023), for each task, we randomly select 500 samples per class from the training set for validation, and use early stopping based on the validation accuracy.

**Baseline Methods for Comparison:** In the experiments, we compare our model with 11 baseline methods including: (1) **Per-task Finetune**, (2) **Continual Finetune** (Wang et al., 2020; Huang et al., 2021), (3) **Prompt Tuning** (Qin & Joty, 2021; Lester et al., 2021), (4) **Data Replay** (Autume et al., 2019), (5) **EWC** (Kirkpatrick et al., 2017), (6) **A-GEM** (Chaudhry et al., 2018), (7) **LFPT5** (Qin & Joty, 2021), (8) **MBPA++** (Autume et al., 2019), (9) **IDBR** (Huang et al., 2021), (10) **Per-task Prompt** (Lester et al., 2021), and (11) **ProgPrompt** (Razdaibiedina et al., 2023). More detailed introductions to these competing methods are provided in Appendix C.3 due to space limitation.

## 4.2 IMPLEMENTATION DETAILS

Q-tuning is a model-backbone-agnostic approach that is applicable to any language models, such as the GPT series (OpenAI, 2023), regardless of their sizes. Due to experimental resource constraints, following (Razdaibiedina et al., 2023), we use two language models including the encoder-decoder T5 model (Raffel et al., 2020) and encoder-only BERT model (Devlin et al., 2018) in our experiments. For all the T5 experiments, we adopt the T5-large model with the text-to-text formulation, where classification labels are mapped into words (e.g. 0/1 will be mapped as "True"/"False"). For all the BERT experiments, we use the BERT-base model as in IDBR and MBPA++ methods (Huang et al., 2021; Autume et al., 2019). Following Devlin et al. (2018), we use the representation of the first token $h_{[CLS]}$ to predict the class of the input text, where $h_{[CLS]}$ is encoded by a beginning-of-a-sentence symbol [CLS]. Following Razdaibiedina et al. (2023), we apply a linear head including a linear transformation parameterized by $\alpha$ and a softmax function to obtain the classification probabilities over classes $k \in \{1...\mathcal{K}\}$: $p(y = k|h) = \frac{\exp(\alpha_k h_{[CLS]})}{\sum_{y \in \mathcal{K}} \exp(\alpha_y h_{[CLS]})}$. The linear head in addition to the prompt embeddings is trained separately for each task. For all the experiments, we set the single prompt length to 10, and apply a parameterized prompt with a two-layer residual MLP[5].

## 5 EXPERIMENTAL RESULTS

We report Q-tuning performance on T5-large and BERT-base models and compare it to previous CL and prompt tuning approaches. We evaluate the methods after training on all tasks and report

---

[3]The details of each order are reported in Table 9 of the Appendix. For each order, as in Razdaibiedina et al. (2023), we train three versions of models, with 16 (or 20), 200, and 1000 training samples per class respectively, and report the performance on the test sets correspondingly.

[4]Please refer to Appendix C.1 and Appendix C.2 for more details.

[5]The rest of the experimental details are reported in Appendix C.3.

253 the averaged test set accuracy across all tasks. The detailed experimental metrics are reported in
254 Appendix C.1. All the experiments are conducted on a single 32GB NVIDIA V100 GPU.

Table 1: Summary of the results with T5 and BERT models on the short-sequence benchmark[6]. Average accuracy after training on the last task is reported. All results are averaged over 3 runs. For T5 experiments, we use few-shot CL settings by following Qin & Joty (2021).

(a) Results with the T5-large model.

| Method | DR | Order 1 | 2 | 3 | avg |
|---|---|---|---|---|---|
| Per-task Finetune | | 70.0 | 70.0 | 70.0 | 70.0 |
| Continual Finetune$^\square$ | | 18.9 | 24.9 | 41.7 | 28.5 |
| Data Replay | ✓ | 35.4 | 37.1 | 41.5 | 38.0 |
| EWC$^\square$ | | 39.0 | 38.0 | 44.8 | 40.6 |
| LFPT5$^{*\square}$ | ✓ | 47.6 | 52.6 | 57.9 | 52.7 |
| ProgPrompt$^*$ | | 74.1 | 74.2 | 75.3 | 74.5 |
| Ours$^*$ | | **75.8** | **75.8** | **76.9** | **76.2** |

(b) Results with the BERT-base model.

| Method | DR | Order 4 | 5 | 6 | 7 | avg |
|---|---|---|---|---|---|---|
| Per-task Finetune | | 73.9 | 73.9 | 73.9 | 73.9 | 73.9 |
| Continual Finetune$^\diamond$ | | 14.8 | 27.8 | 26.7 | 4.5 | 18.4 |
| Data Replay$^\diamond$ | ✓ | 67.2 | 64.7 | 64.7 | 44.6 | 57.8 |
| A-GEM$^\diamond$ | ✓ | 70.6 | 65.9 | 67.5 | 63.6 | 66.9 |
| MBPA++$^\diamond$ | ✓ | 70.8 | 70.9 | 70.2 | 70.7 | 70.6 |
| IDBR$^\dagger$ | ✓ | 75.9 | 76.2 | 76.4 | 76.7 | 76.3 |
| ProgPrompt$^*$ | | 77.8 | 77.5 | 77.6 | 77.4 | 77.6 |
| Ours$^*$ | | **78.5** | **78.3** | **78.3** | **78.4** | **78.4** |

Table 2: Average test set performance of Q-tuning and prior approaches on long-sequence experiments with 15 text classification tasks in different orders. In the experiments[7], we use the few-shot CL by setting 20 samples per class. All the results are averaged over 3 runs.

| Method | | T5-large Order 8 | Order 9 | Order 10 | Average | BERT-base Order 8 | Order 9 | Order 10 | Average |
|---|---|---|---|---|---|---|---|---|---|
| Continual Finetune | | 9.3 | 9.5 | 10.4 | 9.7 | 29.9 | 30.5 | 33.6 | 31.3 |
| Prompt Tuning$^*$ | | 9.7 | 24.4 | 12.2 | 17.4 | - | - | - | - |
| Data Replay | | 46.0 | 50.3 | 34.6 | 43.6 | 34.9 | 39.3 | 34.9 | 36.4 |
| LFPT5$^*$ | | 54.7 | 54.1 | 54.2 | 54.3 | - | - | - | - |
| Per-task Prompt$^*$ | | 69.9 | 69.9 | 69.9 | 69.8 | 50.6 | 50.6 | 50.6 | 50.6 |
| IDBR | | - | - | - | - | 39.7 | 37.9 | 32.9 | 36.8 |
| ProgPrompt$^*$ | | 75.4 | 76.6 | 76.7 | 76.2 | 55.3 | 53.3 | 51.9 | 53.5 |
| Ours$^*$ ($Q_{size} = 5$) | Random | 76.4 | 77.3 | 76.1 | 76.6 | 53.6 | 53.2 | 51.1 | 52.6 |
| | FIFO | 76.5 | 77.2 | 76.7 | 76.8 | 54.5 | 53.8 | 51.8 | 53.4 |
| | DQ-PCA | 77.5 | 78.8 | 77.8 | 78.0 | 55.6 | 56.0 | 51.8 | 54.5 |
| Ours$^*$ ($Q_{size} = 10$) | Random | 76.7 | 77.2 | 76.5 | 76.8 | 54.7 | 54.2 | 52.8 | 53.9 |
| | FIFO | 77.0 | 77.1 | 76.7 | 76.9 | 54.6 | 54.2 | **52.9** | 53.9 |
| | DQ-PCA | **78.3** | **79.7** | **78.7** | **78.9** | **56.5** | **56.2** | 52.6 | **55.1** |
| Ours$^*$ (Full Prompts) | | 79.0 | 79.1 | 78.1 | 78.7 | 55.3 | 55.2 | 54.5 | 55.0 |
| MTL | | 70.7 | 70.7 | 70.7 | 70.7 | 56.9 | 56.9 | 56.9 | 56.9 |

## 5.1 RESULTS ON SHORT-SEQUENCE CL BENCHMARKS

256 Following ProgPrompt (Razdaibiedina et al., 2023), we evaluate the performance of Q-tuning on
257 the standard short-sequence CL benchmarks with few-shot learning settings, where Orders 1∼3 and
258 Orders 4∼7 are evaluated with the T5 and BERT models, respectively. Since these sequential tasks
259 only consist of four or five disjoint datasets, we set $Q_{size} = 5$ for the Q-prompt without utilizing the
260 DQ-PCA rule. In Table 1a, we compare Q-tuning with the existing CL, prompt tuning and continual
261 prompt tuning approaches using the T5 model. Q-tuning outperforms all the CL approaches by a
262 large margin, achieving 76.2% accuracy on average of all the orders. Q-tuning increases the accuracy
263 by 1.7% (from 74.5% to 76.2%) compared to ProgPrompt, the SOTA approach of continual prompt
264 tuning. Q-tuning also surpasses the "Per-task Fintune" by 6.2% on average, demonstrating the efficacy
265 of the proposed queue ensemble and shared prefix prompt approach in enhancing the FKT capability.
266 Table 1b reports the results on the BERT-base model that verify an consistent improvement.

---

[6]Methods marked with $^*$ use soft prompt tuning, while other methods train the entire model. For ProgPrompt, the results are reported by running their released code. DR denotes whether the method requires data replay. $^\square$, $^\diamond$ and $^\dagger$ mark the results from Qin & Joty (2021), Autume et al. (2019) and Huang et al. (2021), respectively.

[7]MTL denotes multi-task learning that fintunes the model using all the datasets from different tasks. Methods marked with $*$ only train a soft prompt while freezing the pretrained model, other methods train the entire model. The "Full Prompts" denotes remaining all prompts in queue by setting $Q_{size} = 15$.

## 5.2 RESULTS ON LONG-SEQUENCE CL BENCHMARKS

In Table 2, we compare the Q-tuning with the baseline approaches on the long-sequence CL bench-mark, including Orders 8~10 using the T5-large and the BERT-base models. These experiments consist of 15 tasks in three different orders. We follow the few-shot CL setting as in Qin & Joty (2021); Razdaibiedina et al. (2023) by selecting 20 samples per class. The row of "Ours (Full Prompts)" denotes the result of not trimming Q-prompt during Q-tuning, *i.e.*, maintaining the complete 15 prompts as in ProgPrompt. As shown in Table 2, the full Q-prompt outperforms ProgPrompt by 2.5% in accuracy on average from 76.2% to 78.7% with the T5 model, which demonstrates again the efficacy of the queue ensemble and shared prefix prompt. Moreover, setting the maximum length of the Q-prompt to 5 using DQ-PCA only leads to a 0.7% accuracy drop (from 78.7% to 78.0%) compared with the full Q-prompt, and we even observe a 0.2% accuracy increase over the full prompt when setting the maximum Q-prompt length to 10. This indicates the capability of DQ-PCA to protect essential knowledge when trimming the Q-prompt. Furthermore, we compare three dequeuing rules to trim the Q-prompt, including random dropping, first in and first out (FIFO), and DQ-PCA. DQ-PCA clearly outperforms the other two naive strategies. We observe consistent improvement in both the T5-large model and the BERT-base model.

Lastly, Table 3 reports the results of Q-tuning on Orders 11~13 including three *random* permutations of 70 *disjoint* tasks, which mimic the lifelong learning scenarios. Training ProgPrompt will fail due to out of memory caused by the accumulation of prompts[8]. Compared to the per-task prompt tuning, Q-tuning has gained considerable performance benefits (30.4% accuracy improvement on average from 60.4% to 90.8%). This can be attributed to 1) the improved FKT by applying Q-prompt ensemble, 2) the effective trimming

Table 3: Results on extremely long sequence experiments (70 randomly permuted tasks). All results are averaged over 3 runs.

| Method | T5-large | | | |
| | Order 11 | Order 12 | Order 13 | **Average** |
| --- | --- | --- | --- | --- |
| ProgPrompt[8] | - | - | - | - |
| Per-task Prompt | 60.4 | 60.4 | 60.4 | 60.4 |
| Shared Prompt | 62.4 | 62.7 | 63.1 | 62.7 |
| Q-tuning ($\mathcal{Q}_{\text{size}} = 10$) | **90.9** | **90.6** | **90.8** | **90.8** |

of Q-prompt using DQ-PCA to enable the training of long sequence of tasks, and 3) the use of shared prefix prompt to avoid the accumulated information loss caused by the Q-prompt trimming. We also compare Q-tuning with training using a global shared prompt and a per-task prompt plus the MR regularization for each task without maintaining the queue of task-specific prompts. To ensure a fair comparison, we set the length of the shared prompt to be identical to Q-tuning, *i.e.*, $l \times \mathcal{Q}_{\text{size}}$. Although the accuracy of the shared prompt is better than the per-task prompt tuning (2.3% improvement on average from 60.4% to 62.7%), it is outperformed by Q-tuning by 28.1% (62.7% to 90.8%) on average. This indicates that, although the Q-prompt and the shared prefix prompt serve the same purpose of aggregating knowledge for better FKT, it is beneficial to keep both components.

Table 4: Forward knowledge transfer results of Order 9 using 20 samples/class. All results are averaged over 3 runs.

| Forward Transfer (Target Task) | Q-prompt (Full) | Q-prompt ($\mathcal{Q}_{\text{size}} = 5$) | Q-prompt ($\mathcal{Q}_{\text{size}} = 10$) | Prompt Tuning |
| --- | --- | --- | --- | --- |
| Task 11 | 98.1 | 97.8 ($\downarrow$0.3%) | 98.2 ($\uparrow$0.1%) | 97.1 ($\downarrow$1.0%) |
| Task 12 | 86.2 | 83.9 ($\downarrow$2.3%) | 86.1 ($\downarrow$0.1%) | 72.6 ($\downarrow$13.6%) |
| Task 13 | 56.6 | 54.9 ($\downarrow$1.7%) | 56.2 ($\downarrow$0.4%) | 49.8 ($\downarrow$6.8%) |
| Task 14 | 50.4 | 50.3 ($\downarrow$0.1%) | 50.5 ($\uparrow$0.1%) | 47.6 ($\downarrow$2.8%) |
| Task 15 | 69.4 | 68.9 ($\downarrow$0.5%) | 69.1 ($\downarrow$0.3%) | 68.1 ($\downarrow$1.3%) |
| Average | 72.1 | 71.2 ($\downarrow$0.9%) | **72.0** ($\downarrow$0.1%) | 67.0 ($\downarrow$5.1%) |

Table 5: Ablation studies on the Q-prompt ensemble and prefix shared prompt of Q-tuning[9]. All results are averaged over 3 runs.

| Sequence | Method | | | Num. samples | | | |
| | Q-prompt | Ensemble | $\theta_{\mathcal{P}*}$ | 16 | 200 | 1000 | **Average** |
| --- | --- | --- | --- | --- | --- | --- | --- |
| Short | ✓ | ✗ | ✗ | 74.5 | 79.8 | 79.8 | 78.0 |
| | ✓ | ✓ | ✗ | 75.2 | 80.9 | 80.4 | 78.8 |
| | ✓ | ✗ | ✓ | 75.1 | 80.6 | 80.9 | 78.9 |
| | ✓ | ✓ | ✓ | **76.2** | **81.2** | **81.9** | **79.7** |

| Sequence | Method | | | Num. samples | | | |
| | Q-prompt | Ensemble | $\theta_{\mathcal{P}*}$ | 20 | 200 | 1000 | **Average** |
| --- | --- | --- | --- | --- | --- | --- | --- |
| Long | ✓ | ✗ | ✗ | 76.7 | 80.8 | 80.8 | 79.4 |
| | ✓ | ✓ | ✗ | 77.2 | 81.1 | 82.1 | 80.2 |
| | ✓ | ✗ | ✓ | 77.4 | 81.1 | 82.3 | 80.3 |
| | ✓ | ✓ | ✓ | **78.9** | **81.9** | **83.3** | **81.4** |

## 5.3 ABLATION STUDY AND ANALYSIS

In this section, we evaluate our approach's performances in various aspects, including its capability of fulfilling FKT, adapting previous prompts based on their relevance to the new task using the Q-prompt ensemble, and maintaining global knowledge sharing using a shared prefix prompt.

**Forward Knowledge Transfer:** In Table 4, we evaluate the FKT performance of the trimmed Q-prompt. We train three different Q-prompts including the "Full", "$\mathcal{Q}_{\text{size}} = 5$" and "$\mathcal{Q}_{\text{size}} = 10$",

---

[8]In our experiments, training ProgPrompt fails after the 15-th task on a single NVIDIA V100 GPU (32GB)

[9]For long sequence, we set $\mathcal{Q}_{\text{size}} = 10$. More detailed results of each order are reported in Appendix D.

where the "Full" denotes keeping the complete Q-prompt without the De-Q operation. All these Q-prompts are continuously trained on the first 10 tasks of Order 9. Then we separately evaluate the FKT performance of these Q-prompts on five remaining target tasks. As a reference, we also train a single prompt (denoted by "Prompt Tuning" whose token length is set the same as the total length of the full Q-prompt) on each target task. First of all, full Q-prompt substantially outperforms "Prompt Tuning", demonstrating our approach's capability in fulfilling FKT whereas "Prompt Tuning" does not leverage any information from other tasks. Moreover, compared to the full Q-prompt, the trimmed Q-prompt only has a minor performance drop. For example, setting $\mathcal{Q}_{\text{size}} = 10$ only leads to 0.1% accuracy decrease (from 72.1% to 72.0%). This proves that trimmed Q-prompt is able to maintain FKT at the same level as the full Q-prompt, despite previous prompts being trimmed.

**Q-prompt Ensemble:** Table 5 demonstrates the efficacy of the Q-prompt ensemble. In both the short and long task sequences, compared with the complete Q-prompt model (the fourth row), dropping the ensemble (the third row) leads to 0.8% and 1.1% accuracy drop in the short and long task sequences, respectively. In addition, in Fig. 2, we visualize the trained weight matrix $\mathcal{W}$ to reflect the relevance of previously learned prompts to the current task. We can observe when learning the "sst2" task, the prompt from the "imdb" task contributes the most. This is because the two tasks are both for the movie review classification. The ensemble matrix uncovers their correlation and assigns more weights to the prompt of the "imdb" task. In contrast, for the "qnli" task, the ensemble matrix suggests an even contribution of each prompt in the queue. This is because all the tasks are related to the Q&A classification.

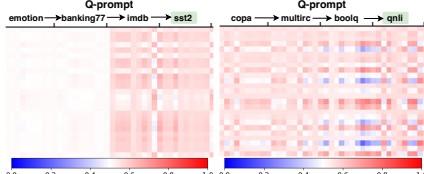

Figure 2: Visualization of ensemble matrix.

Table 6: Ablation studies on the extremely long sequence experiments. All results are averaged over 3 runs.

| Method | | T5-large | | | |
|---|---|---|---|---|---|
| $\theta_{\mathcal{P}*}$ | $\mathcal{L}_{\text{MR}}$ | Order 11 | Order 12 | Order 13 | **Average** |
| ✗ | ✗ | 86.8 | 87.3 | 87.7 | 87.3 |
| ✓ | ✗ | 89.8 | 89.4 | 90.1 | 89.8 |
| ✓ | ✓ | **90.9** | **90.6** | **90.8** | **90.8** |

**Shared Prefix Prompt:** We conduct ablation studies to validate the efficacy of the shared prefix prompt. As shown in Table 5, in both the short and long task sequences, by comparing the complete Q-prompt model (the fourth row) and dropping the shared prefix prompt (the second row), we observe an accuracy drop of 0.9% and 1.2% in the short and long task sequences, respectively. The impact in the short task sequence is less than that of the long task sequence. This is expected as the short task sequence does not utilize DQ-PCA to trim the Q-prompt, hence no information loss from previous prompts. This will dilute the effect of the shared prefix prompt. Furthermore, to evaluate the contribution of the MR regularization, we conduct the experiments on a long task sequence by setting $\mathcal{Q}_{\text{size}} = 10$. As shown in Table 6, dropping the MR regularization from the shared prefix prompt (from the third row to the second row) leads to a 1% accuracy drop. We also evaluate the performance using different $\eta$ values for the MR regularization, which is reported in Appendix D.

## 6 CONCLUSION

This paper introduces a new model-agnostic approach named Q-tuning, which can pave the way to achieving lifelong continual prompt tuning for present and future LMs with a rapid growth of parameters. In comparison with existing CL methods, Q-tuning maintains a low-cost prompt queue instead of storing a large number of task-specific parameters or saving old data samples for replay. Our extensive experiments demonstrate that Q-tuning outperforms existing continual learning, prompt tuning and continual prompt tuning methods on the standard CL benchmarks for text classification. In addition, we verify the effectiveness of Q-tuning on both short and long task sequences, including up to 70 tasks that mimic the case of lifelong learning.

**Limitations:** Although Q-tuning demonstrates a strong FKT capability, it does not enable the backward knowledge transfer as both the model and the previous Q-prompts are frozen during the learning of a new task. Besides, Q-tuning requires the task identities to be known at test time. To address the more challenging CL scenario when the task identities are undisclosed at test time, inspired by Wang et al. (2022), for task $i$, we can assign a trainable query key $k^i$ to the corresponding Q-prompt $\mathcal{Q}^i$ and jointly train $k^i$ to maximize the similarity between $k^i$ and the feature of each sample $x$ from task $i$. During test time, given an input $x'$ with an unknown identity, we will first locate the Q-prompt that has the largest similarity between its key $k^j$ and the input $x'$, and then we can use the retrieved Q-prompt $\mathcal{Q}^j$ to infer $x'$. We will address this problem in our future work.

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

# Appendix

## A   Q-TUNING ALGORITHM

---

**Algorithm 1** Q-tuning Algorithm

---

**Input:** Continually increased task set $\mathcal{T}$, Q-prompt $\mathcal{Q}$ with a maximum capacity $C$, fixed pretrained
   model $\theta_{\mathcal{M}}$, ensemble matrix $\mathcal{W}$ for $\mathcal{Q}$, shared prefix prompt $\theta_{\mathcal{P}*}$, memory factor $\eta$.
**Initialize:** $\mathcal{Q}^1 = \{\}$, randomly initialized $\theta_{\mathcal{P}*}^1$ and $\theta_{\mathcal{P}}^1$, initialized $\mathcal{W}^1$ with an identity matrix.
   **for** continually coming task $i = 1, 2, \ldots$ **do**
      **if** $i > C$ **then**
         $\mathcal{Q} \leftarrow$ PCA-DQ$(\mathcal{Q})$ // De-Q (Eq.(4))
      **else**
         $\mathcal{Q} \leftarrow \mathcal{Q} \oplus \theta_{\mathcal{P}}^i$ // En-Q
      **end if**
      **for** each batch sample from $\mathcal{T}^i$'s dataset **do**
         $\theta_{\mathcal{P}}^i \leftarrow \theta_{\mathcal{P}}^i + \nabla_{\theta_{\mathcal{P}}^i} \mathcal{L}_{\mathcal{Q}}(\theta_{\mathcal{P}*}^i, \theta_{\mathcal{P}}^i, \mathcal{W}^i)$
         $\mathcal{W}^i \leftarrow \mathcal{W}^i + \nabla_{\mathcal{W}^i} \mathcal{L}_{\mathcal{Q}}(\theta_{\mathcal{P}*}^i, \theta_{\mathcal{P}}^i, \mathcal{W}^i)$
         **if** i=1 **then**
            $\theta_{\mathcal{P}*}^i \leftarrow \theta_{\mathcal{P}*}^i + \nabla_{\theta_{\mathcal{P}*}^i} \mathcal{L}_{\mathcal{Q}}(\theta_{\mathcal{P}*}^i, \theta_{\mathcal{P}}^i, \mathcal{W}^i)$
         **else if** $i > C$ **then**
            $\theta_{\mathcal{P}*}^i \leftarrow \theta_{\mathcal{P}*}^i + \nabla_{\theta_{\mathcal{P}*}^i} [\mathcal{L}_{\mathcal{Q}}(\theta_{\mathcal{P}*}^i, \theta_{\mathcal{P}}^i, \mathcal{W}^i) + \eta \mathcal{L}_{\mathrm{MR}}(\theta_{\mathcal{P}*}^i)]$
         **end if**
      **end for**
   **end for**

---

## B   MUTUAL INFORMATION ESTIMATION

**Proposition 1.** *Let $p(x)$ and $p(y)$ represent two random variables, their mutual information satisfies*

$$
\begin{aligned}
I(p(x); p(y)) &:= \mathcal{D}_{\mathrm{JSD}}(\mathbf{J}||\mathbf{M}) \\
&\geq \mathbf{E}_{z \sim \mathbf{J}}\left[-\sigma(-\mathcal{F}_\omega(z))\right] - \mathbf{E}_{z' \sim \mathbf{M}}\left[\sigma(\mathcal{F}_\omega(z'))\right]
\end{aligned}
\tag{10}
$$

*where the joint $\mathbf{J} = p(x,y)$, $\mathbf{M} = p(x)p(y)$ is the product of the marginals, $\sigma(t) = \log(1 + e^t)$, and $\mathcal{F}$ belongs to an arbitrary class of functions that can map $\mathbf{J} \to \mathbb{R}$ and $\mathbf{M} \to \mathbb{R}$.*

*Proof.* According to the variational estimation of $f$-divergences (Nguyen et al., 2010), we have

$$
\begin{aligned}
\mathcal{D}_f(\mathbf{P}||\mathbf{Q}) &= \int q(x) \sup_{t \in \mathrm{dom}_{g^*}} t\frac{p(x)}{q(x)} - g^*(t)\mathrm{d}x \\
&\geq \sup_{\mathcal{V} \in F}\left(\int p(x)\mathcal{V}(x)\mathrm{d}x - \int q(x)g^*(\mathcal{V}(x))\mathrm{d}x\right) \\
&= \sup_{\mathcal{V} \in F}(\mathbb{E}_{x \sim \mathbf{P}}[\mathcal{V}(x)] - \mathbb{E}_{x \sim \mathbf{Q}}[g^*(\mathcal{V}(x))])
\end{aligned}
\tag{11}
$$

where the function $g^*$ is a convex conjugate function (Hiriart-Urruty & Lemaréchal, 2004; Nowozin et al., 2016) of a convex, lower-semicontinuous function. The function $g^*$ is defined as

$$
g^*(t) = \sup_{u \in \mathrm{dom}_f}\{ut - f(u)\}
\tag{12}
$$

We parameterize $\mathcal{V}$ using a neural network with parameters $w$ and write it as $\mathcal{V}_\omega$. We assume the form of the function $\mathcal{V}_\omega = g_f(\mathcal{F}_\omega(x))$. Given two probability distributions $\mathbf{J} = p(x,y)$ and $\mathbf{M} = p(x)p(y)$, their $f$-divergence satisfies:

$$
\mathcal{D}_f(\mathbf{J}||\mathbf{M}) = \sup_{\mathcal{F}_\omega}(\mathbb{E}_{z \sim \mathbf{J}}[g_f(\mathcal{F}_\omega(z))] - \mathbb{E}_{z' \sim \mathbf{M}}[g^*(g_f(\mathcal{F}_\omega(z')))])
\tag{13}
$$

Table 7: Recommended final layer activation functions and their conjugate functions. This table comes from Nowozin et al. (2016).

| Name | Output activation $g_f$ | $\mathbf{dom}_{g^\star}$ | Conjugate $g^\star(t)$ |
|---|---|---|---|
| Kullback-Leibler (KL) | $v$ | $\mathbb{R}$ | $\exp(t-1)$ |
| Reverse KL | $-\exp(-v)$ | $\mathbb{R}_-$ | $-1-\log(-t)$ |
| Pearson $\chi^2$ | $v$ | $\mathbb{R}$ | $\frac{1}{4}t^2 + t$ |
| Square Hellinger | $1 - \exp(-v)$ | $t < 1$ | $\frac{t}{1-t}$ |
| Jensen-Shannon | $\log(2) - \log(1 + \exp(-v))$ | $t < \log(2)$ | $-\log(2 - \exp(t))$ |

where $g_f$ is an activation function specific to the $f$-divergence used. Table 7 provides the commonly used $g_f$ and the convex conjugate function $g^*$. According to this table, for the JSD based divergence, we have $g_f(x) = \log(2) - \log(1 + \exp(-x))$ and $g^*(x) = -\log(2 - \exp(x))$. By substituting them into Eq. (13), we have

$$
\begin{aligned}
\mathbb{E}_{z\sim\mathbf{J}}\left[g_f(\mathcal{F}_\omega(z))\right] &= \mathbb{E}\left[\log 2 - \log(1 + \exp(-\mathcal{F}_\omega(z)))\right] \\
&= \mathbb{E}_{z\sim\mathbf{J}}\left[\log 2 - \sigma(-\mathcal{F}_\omega(z))\right]
\end{aligned}
\tag{14}
$$

$$
\begin{aligned}
&\mathbb{E}_{z'\sim\mathbf{M}}\left[g^*(g_f(\mathcal{F}_\omega(z')))\right] \\
&= \mathbb{E}_{z'\sim\mathbf{M}}\left[-\log(2 - \exp^{\log 2 - \log(1 + \exp(-\mathcal{F}_\omega(z')))})\right] \\
&= \mathbb{E}_{z'\sim\mathbf{M}}\left[-\log(2 - 2(1 + \exp(-\mathcal{F}_\omega(z'))^{-1}))\right] \\
&= \mathbb{E}_{z'\sim\mathbf{M}}\left[-\log(2\frac{\exp(-\mathcal{F}_\omega(z'))}{1 + \exp(-\mathcal{F}_\omega(z'))})\right] \\
&= \mathbb{E}_{z'\sim\mathbf{M}}\left[-\log\frac{2\exp(-\mathcal{F}_\omega(z'))\exp(\mathcal{F}_\omega(z'))}{\exp(\mathcal{F}_\omega(z')) + \exp(-\mathcal{F}_\omega(z'))\exp(\mathcal{F}_\omega(z'))}\right] \\
&= \mathbb{E}_{z'\sim\mathbf{M}}\left[-\log(\frac{2}{\exp(\mathcal{F}_\omega(z')) + 1})\right] \\
&= \mathbb{E}_{z'\sim\mathbf{M}}\left[-(\log 2 - \log(\exp(\mathcal{F}_\omega(z')) + 1))\right] \\
&= \mathbb{E}_{z'\sim\mathbf{M}}\left[-\log 2 + \sigma(\mathcal{F}_\omega(z'))\right]
\end{aligned}
\tag{15}
$$

Combining Eq. (14) and Eq. (15), we can rewrite Eq. (13) as a JSD-divergence based form:

$$
\begin{aligned}
\mathcal{D}_{\text{JSD}}(\mathbf{J}||\mathbf{M}) = \sup_{\mathcal{F}_\omega}(&\mathbb{E}_{z\sim\mathbf{J}}\left[\log 2\right] + \mathbb{E}_{z\sim\mathbf{J}}\left[-\sigma(-\mathcal{F}_\omega(z))\right] \\
&+ \mathbb{E}_{z'\sim\mathbf{M}}\left[\log 2\right] - \mathbb{E}_{z'\sim\mathbf{M}}\left[\sigma(\mathcal{F}_\omega(z'))\right]) \\
\geq &\mathbb{E}_{z\sim\mathbf{J}}\left[-\sigma(-\mathcal{F}_\omega(z))\right] - \mathbb{E}_{z'\sim\mathbf{M}}\left[\sigma(\mathcal{F}_\omega(z'))\right]
\end{aligned}
\tag{16}
$$

$\square$

## C  FURTHER IMPLEMENTATION DETAILS

### C.1  DATASETS AND METRICS

We use 21 public datasets, of which 15 datesets are the same as ProgPrompt Razdaibiedina et al. (2023) for our experiments. Table 8 reports the details of the 21 datasets, along with their evaluation metrics. Overall, we use datasets from CL benchmark (Zhang et al., 2015), GLUE (Wang et al., 2018) and SuperGLUE (Wang et al., 2019) benchmarks, and IMDB movie reviews dataset. We use the Banking77 dataset (Casanueva et al., 2020) and Emotion dataset (Saravia et al., 2018) for the extremely long 70-task experiments. Following the common practice, for tasks that have two evaluation metrics, we use the average of the two as the final performance metric.

To mimic the life-long learning, we add WNLI, COLA and QNLI from the GLUE benchmark, WSC from the SuperGLUE benchmark, the Banking77 dataset (Casanueva et al., 2020) and the Emotion dataset (Saravia et al., 2018) to form an extremely long sequence including 70 tasks. In the 70-task

experiments, we split the DBpedia set into 7 **disjoint** tasks, the Yahoo set into 5 **disjoint** tasks, and the Banking77 set into 38 **disjoint** tasks (removing 1 class), and the Emotion dataset into 3 **disjoint** tasks, where each task has two 2 classes. These divided 53 subsets plus the rest 17 datasets form the final 70-task dataset. Following Razdaibiedina et al. (2023), for each task, we randomly select 500 samples per class from the training set for validation, and use early stopping according to the validation accuracy on all seen tasks.

Table 8: The details of 21 datasets used in our experiments. NLI denotes natural language inference, QA denotes questions and answers task, and EM denotes exact match scoring. The first five tasks are used to form the standard CL benchmark, all other tasks are used in our long-sequence experiments.

| Dataset name | Category | Task | Domain | Metric | Classes |
|---|---|---|---|---|---|
| 1. YP | CL benchmark | sentiment analysis | YP reviews | accuracy | 5 |
| 2. Amazon | CL benchmark | sentiment analysis | Amazon reviews | accuracy | 5 |
| 3. DBpedia | CL benchmark | topic classification | Wikipedia | accuracy | 14 |
| 4. Yahoo | CL benchmark | QA | Yahoo Q&A | accuracy | 10 |
| 5. AG News | CL benchmark | topic classification | news | accuracy | 4 |
| 6. MNLI | GLUE | NLI | various | accuracy | 3 |
| 7. QQP | GLUE | paraphrase detection | Quora | accuracy & F1 | 2 |
| 8. RTE | GLUE | NLI | news, Wikipedia | accuracy | 2 |
| 9. SST2 | GLUE | sentiment analysis | movie reviews | accuracy | 2 |
| 10. WiC | SuperGLUE | word sense disambiguation | lexical databases | accuracy | 2 |
| 11. CB | SuperGLUE | NLI | various | accuracy | 2 |
| 12. COPA | SuperGLUE | QA | blogs, encyclopedia | accuracy | 2 |
| 13. BoolQ | SuperGLUE | boolean QA | Wikipedia | accuracy | 2 |
| 14. MultiRC | SuperGLUE | QA | various | F1 & EM | 2 |
| 15. IMDB | Other | sentiment analysis | movie reviews | accuracy | 2 |
| 16. WNLI | GLUE | NLI | various | accuracy | 2 |
| 17. COLA | GLUE | NLI | books, journal articles | accuracy | 2 |
| 18. QNLI | GLUE | QA | Wikipedia | accuracy | 2 |
| 19. WSC | SuperGLUE | NLI | various | accuracy | 2 |
| 20. Banking77 | Other | intent detection | banking | accuracy | 77 |
| 21. Emotion | Other | emotion detection | Twitter | accuracy | 6 |

## C.2 TASK SEQUENCE ORDERS

We report the task orders used in our experiments across the T5 and BERT models in Table 9 below, where Orders 1-10 are the same as ProgPrompt (Razdaibiedina et al., 2023). The Orders 11-13 are created by **randomly permuting** the collected 70 disjoint datasets to mimic the lifelong learning of continuously incoming unseen tasks.

Table 9: Thirteen different orders of task sequences used for continual learning experiments. Orders 1-7 correspond to the standard CL benchmarks adopted by prior works (Razdaibiedina et al., 2023) for short-sequence experiments. Orders 8-10 are long-sequence orders spanning 15 tasks. Orders 11-13 are our customized extremely long sequences, where the tasks are **randomly permuted**. In these extremely long cases, existing techniques such as the SOTA, ProgPrompt (Razdaibiedina et al., 2023), cannot cope with these long tasks, due to the quadratic growing training and inference costs.

| Order | Model | Task Sequence |
|---|---|---|
| 1 | T5 | db → amazon → yahoo → ag |
| 2 | T5 | db → amazon → ag → yahoo |
| 3 | T5 | yahoo → amazon → ag → db |
| 4 | BERT | ag → yp → amazon → yahoo → db |
| 5 | BERT | yp → yahoo → amazon → db → ag |
| 6 | BERT | db → yahoo → ag → amazon → yp |
| 7 | BERT | yp → ag → db → amazon → yahoo |
| 8 | T5, BERT | mnli → cb → wic → copa → qqp → boolq → rte → imdb → yp → amazon → sst2 → dbpedia → ag → multirc → yahoo |
| 9 | T5, BERT | multirc → boolq → wic → mnli → cb → copa → qqp → rte → imdb → sst2 → dbpedia → ag → yp → amazon → yahoo |
| 10 | T5, BERT | yp → amazon → mnli → cb → copa → qqp → rte → imdb → sst2 → dbpedia → ag → yahoo → multirc → boolq → wic |
| 11 | T5 | wsc → banking77-19 → banking77-9 → banking77-8 → banking77-25 → yahoo-1 → banking77-34 → banking77-3 → banking77-23 → cb → banking77-7 → banking77-35 → banking77-13 → imdb → banking77-12 → banking77-17 → multirc → banking77-14 → emotion-0 → banking77-22 → yp → dbpedia-14-5 → banking77-30 → banking77-1 → banking77-15 → boolq → banking77-20 → banking77-21 → dbpedia-14-2 → qnli → banking77-31 → banking77-29 → emotion-2 → yahoo-3 → dbpedia-14-1 → banking77-32 → banking77-0 → rte → ag-news → dbpedia-14-4 → banking77-2 → yahoo-4 → banking77-11 → banking77-37 → banking77-27 → sst2 → banking77-33 → copa → banking77-5 → dbpedia-14-0 → wic → qqp → banking77-26 → yahoo-2 → banking77-10 → banking77-36 → banking77-4 → emotion-1 → dbpedia-14-3 → amazon → banking77-28 → banking77-16 → banking77-24 → mnli → cola → wnli → banking77-18 → banking77-6 → dbpedia-14-6 → yahoo-0 |
| 12 | T5 | banking77-29 → yp → banking77-30 → banking77-26 → banking77-20 → yahoo-2 → amazon → dbpedia-14-2 → banking77-24 → yahoo-3 → banking77-22 → banking77-16 → yahoo-0 → dbpedia-14-1 → emotion-2 → dbpedia-14-4 → dbpedia-14-6 → wic → banking77-23 → banking77-14 → banking77-18 → yahoo-4 → banking77-5 → banking77-0 → banking77-13 → cb → banking77-35 → rte → banking77-4 → dbpedia-14-3 → banking77-1 → banking77-9 → banking77-15 → banking77-3 → banking77-6 → banking77-21 → mnli → banking77-2 → yahoo-1 → boolq → banking77-10 → banking77-25 → banking77-37 → banking77-17 → qqp → banking77-28 → wnli → banking77-8 → banking77-31 → dbpedia-14-0 → banking77-11 → banking77-27 → banking77-7 → multirc → banking77-33 → banking77-12 → imdb → copa → banking77-19 → cola → banking77-34 → sst2 → emotion-0 → wsc → qnli → emotion-1 → banking77-32 → dbpedia-14-5 → ag-news → banking77-36 |
| 13 | T5 | yahoo-2 → copa → banking77-22 → emotion-0 → banking77-1 → emotion-1 → yahoo-0 → banking77-32 → banking77-37 → dbpedia-14-0 → banking77-3 → qnli → multirc → banking77-0 → dbpedia-14-3 → ag-news → banking77-10 → imdb → banking77-5 → banking77-15 → banking77-16 → wnli → banking77-36 → wsc → banking77-13 → banking77-19 → amazon → banking77-29 → banking77-33 → boolq → banking77-28 → yahoo-1 → yp → banking77-14 → emotion-2 → mnli → banking77-7 → banking77-21 → banking77-30 → banking77-4 → banking77-9 → banking77-35 → dbpedia-14-5 → banking77-26 → cola → qqp → yahoo-3 → dbpedia-14-6 → wic → banking77-25 → banking77-31 → banking77-17 → banking77-23 → banking77-8 → cb → banking77-6 → dbpedia-14-2 → banking77-20 → dbpedia-14-1 → yahoo-4 → banking77-18 → banking77-2 → banking77-34 → banking77-12 → dbpedia-14-4 → banking77-27 → rte → sst2 → banking77-24 → banking77-11 |

## C.3 IMPLEMENTATION AND EXPERIMENT DETAILS

**More Details of the Methods for Comparison** Following Razdaibiedina et al. (2023), we consider 11 baseline methods for comparison with the proposed Q-tuning:

- **Per-task Finetune** separately tunes the whole model for each task. We use this type of method as a baseline in the short-sequence benchmark experiments.

- **Continual Finetune** (Wang et al., 2020; Huang et al., 2021) continually tunes the whole model on a sequence of tasks without adding any regularization or replaying data from the previous tasks.

- **Prompt Tuning** (Qin & Joty, 2021; Lester et al., 2021) sequentially trains a shared soft prompt across all tasks, while freezing the pretrained model.

- **Data Replay** finetunes the whole model for new tasks while replaying samples from previous tasks to prevent the CF problem.

- **EWC** (Kirkpatrick et al., 2017) finetunes the whole model using a regularization loss which penalizes updating parameters that could disturb the previously learned tasks.

- **A-GEM** (Chaudhry et al., 2018) retrieves examples from old tasks and restricts the gradients to update the model when learning new tasks.

- **LFPT5** (Qin & Joty, 2021) continuously trains a soft prompt that learns the tasks while generating samples for experience replay.

- **MBPA++** (Autume et al., 2019) uses an episodic memory to augment BERT by storing all seen examples.

- **IDBR** (Huang et al., 2021) continuously trains the whole model by using data replay and a regularization loss. It adopts sentence representation disentanglement in task-specific and task-generic spaces, achieving SOTA on the CL benchmark with BERT.

- **Per-task Prompt** (Lester et al., 2021) trains a separate soft prompt for each task while keeping the original model frozen. This type of method naturally eliminates the CF problem, because separately tuned prompts will not change when new tasks are learned. However, this independent prompt tuning setup cannot achieve forward knowledge transfer.

- **ProgPrompt** (Razdaibiedina et al., 2023) trains a progressively increased prompt list to achieve the forward knowledge transfer and resist the CF problem using prompt tuning without relying on data replay. Current SOTA on continual prompt tuning benchmarks with T5 and BERT.

**Implementation Details** We use PyTorch and HuggingFace Transformers library for our implementation. For the standard CL benchmark, we use official datasets provided by Zhang et al. (2015), following Autume et al. (2019); Zhang et al. (2015). We use HuggingFace datasets (`https://github.com/huggingface/datasets`) to download data for GLUE tasks (Wang et al., 2018), SuperGLUE tasks (Wang et al., 2019) tasks, IMDB movie reviews dataset (Maas et al., 2011), Banking77 dataset (Casanueva et al., 2020), and Emotion dataset (Saravia et al., 2018), which we use for long-sequence CL experiments, life-long learning experiments and ablation studies. Following previous studies (Autume et al., 2019; Razdaibiedina et al., 2023), for CL experiments, for each dataset, we use the available validation set as a test set (since test data is not available), and hold out 500 samples from the train set to construct the validation set. For our ablation studies, we report the maximal validation set performance.

We use the Adam optimizer and set the batch size to 8 for all the experiments. Following Razdaibiedina et al. (2023), we train each prompt between 20 and 300 epochs, depending on the number of data points. We use the prompt checkpoints with the best validation set score as our final prompts. Prompts are initialized from randomly sampled tokens as in Lester et al. (2021), hyperparameters are shown in the Table 10.

The mutual information maximization can be approximated by maximizing its variational lower bound (Barber & Agakov, 2004; Poole et al., 2019) defined by Eq. (6). But this variational approximation requires extra costly computation to optimize the discriminator $\mathcal{F}_w$. We empirically find a KL-divergence based loss can go for the same goal, which is also verified by Müller et al. (2019); Tian et al. (2019). The KL-divergence based MR loss between the new memory and the old memory is

defined as follows:

$$\mathcal{L}_{\text{MR}} = \sum_{i \in |\mathcal{T}|} \sum_{(\mathbf{x}^i, \mathbf{y}^i) \in \mathcal{T}^i} D_{\text{KL}}(p(\mathbf{y}^i | \mathbf{x}^i; \theta_{\mathcal{M}}, \theta_{\mathcal{P}*}^i) \| p(\mathbf{y}^i | \mathbf{x}^i; \theta_{\mathcal{M}}, \mathcal{W}^{i-1} \circ [\theta_{\mathcal{P}*}^{i-1}, \mathcal{Q}^{i-1}])), \quad (17)$$

where only the shared prefix prompt $\theta_{\mathcal{P}*}^i$ is trainable. This MR regularization loss does not require training an extra discriminator network, achieving the same objective as knowledge distillation (Hinton et al., 2015).

For all the CL experiments, we use early stopping as in Huang et al. (2021), to save model checkpoint based on the best validation performance on the current task. We report test set performance after training on all tasks as our final metric. For SuperGLUE experiments, we report maximal validation set performance over the course of training as in Lester et al. (2021). We measure the validation performance after every epoch and use metrics described in Appendix C.1. We use the same hyperparameter settings for all prompt-based approaches (Q-tuning, Progressive Prompts, per-task prompt) as in Razdaibiedina et al. (2023).

Table 10: Hyperparameters used for Q-tuning across different CL experiments.

| Hyperparameter ↓ | Short-sequence benchmark | | | Long-sequence benchmark | | |
|---|---|---|---|---|---|---|
| Num. samples → | 16 | 200 | 1000 | 20 | 200 | 1000 |
| T5-large Model | | | | | | |
| Epochs | 300 | 150 | 20 | 300 | 150 | 20 |
| Learning rate | 0.3 | 0.3 | 0.3 | 0.3 | 0.3 | 0.3 |
| Length of shared prompt $\theta_{\mathcal{P}*}$ | 10 | 10 | 10 | 10 | 10 | 10 |
| Length of each prompt in $\mathcal{Q}$ | 10 | 10 | 10 | 10 | 10 | 10 |
| Memory factor $\eta$ | 0.001 | 0.001 | 0.001 | 0.01 | 0.01 | 0.01 |
| BERT-base Model | | | | | | |
| Epochs | 300 | 150 | 40 | 300 | 150 | 40 |
| Learning rate | 0.0001 | 0.0001 | 0.0001 | 0.0001 | 0.0001 | 0.0001 |
| Length of shared prompt $\theta_{\mathcal{P}*}$ | 10 | 10 | 10 | 5 | 5 | 5 |
| Length of each prompt in $\mathcal{Q}$ | 10 | 10 | 10 | 5 | 5 | 5 |
| Memory factor $\eta$ | 0.001 | 0.001 | 0.001 | 0.01 | 0.01 | 0.01 |

Table 11: More details of the ablation study results on each order reported in Table 5. For the long-sequence experiments, we set the queue size to 10. All results are averaged over 3 runs.

| Sequence | Method | | | T5-large Results | | | | | | | | | | | |
|---|---|---|---|---|---|---|---|---|---|---|---|---|---|---|---|
| | Q-prompt | Ensemble | $\theta_{\mathcal{P}*}$ | Order1 | | | Order2 | | | Order3 | | | Average | | |
| | (Num. samples →) | | | 16 | 200 | 1000 | 16 | 200 | 1000 | 16 | 200 | 1000 | 16 | 200 | 1000 |
| Short | ✓ | | | 74.1 | 80.0 | 79.6 | 74.2 | 79.5 | 79.9 | 75.3 | 79.8 | 80.1 | 74.5 | 79.8 | 79.8 |
| | ✓ | ✓ | | 74.9 | 80.9 | 80.4 | 75.1 | 80.6 | 80.1 | 75.6 | 81.1 | 80.8 | 75.2 | 80.9 | 80.4 |
| | ✓ | | ✓ | 75.0 | 80.7 | 81.6 | 74.6 | 80.7 | 80.7 | 75.7 | 80.4 | 80.6 | 75.1 | 80.6 | 80.9 |
| | ✓ | ✓ | ✓ | 75.8 | 81.2 | 82.3 | 75.8 | 81.1 | 82.2 | 76.9 | 81.1 | 81.1 | **76.2** | **81.2** | **81.9** |

| Sequence | Method | | | T5-large Results | | | | | | | | | | | |
|---|---|---|---|---|---|---|---|---|---|---|---|---|---|---|---|
| | Q-prompt | Ensemble | $\theta_{\mathcal{P}*}$ | Order8 | | | Order9 | | | Order10 | | | Average | | |
| | (Num. samples →) | | | 20 | 200 | 1000 | 20 | 200 | 1000 | 20 | 200 | 1000 | 20 | 200 | 1000 |
| Long | ✓ | | | 76.3 | 81.6 | 81.0 | 76.9 | 80.6 | 80.5 | 76.7 | 80.1 | 80.9 | 76.7 | 80.8 | 80.8 |
| | ✓ | ✓ | | 77.1 | 81.6 | 82.1 | 77.4 | 81.7 | 81.9 | 77.2 | 80.2 | 82.4 | 77.2 | 81.1 | 82.1 |
| | ✓ | | ✓ | 77.4 | 81.7 | 82.5 | 77.9 | 80.9 | 82.5 | 77.1 | 80.7 | 82.0 | 77.4 | 81.1 | 82.3 |
| | ✓ | ✓ | ✓ | 78.3 | 82.4 | 83.5 | 79.7 | 82.1 | 83.3 | 78.7 | 81.4 | 83.1 | **78.9** | **81.9** | **83.3** |

**MLP-based prompt** We follow Razdaibiedina et al. (2023) by setting a two-layer MLP for parameterizing the soft-prompt. The two-layer MLP includes two linear layers with the ReLU activation function. The number of hidden nodes in the hidden layer is set to 512 in all Q-tuning experiments.

## D    MORE ABLATION STUDY RESULTS

Table 11 reports more details of the results on each order in Table 5 for the ablation study. Table 12 presents the effectiveness of setting different memory factors $\eta$ in the MR loss. As shown, the $\eta$ is suggested to $10^{-2}$ for the long sequence tasks. By comparing with the results of "w/o MR", the performance by using MR loss is improved by 1.7% on average.

Table 12: Ablation study experiments (20 samples/class for long sequence) on the memory factor $\eta$ of the MR loss. All results are averaged over 3 runs.

| Parameter | Long Sequence | | | |
| --- | --- | --- | --- | --- |
| | Order 8 | Order 9 | Order 10 | Average |
| $\eta = 1$ | 73.5 | 75.8 | 73.2 | 74.2 |
| $\eta = 10^{-1}$ | 77.1 | 78.6 | 77.3 | 77.7 |
| $\eta = 10^{-2}$ | **78.3** | **79.7** | **78.7** | **78.9** |
| $\eta = 10^{-3}$ | 78.1 | 79.4 | 78.0 | 78.5 |
| $\eta = 10^{-4}$ | 77.8 | 78.8 | 77.8 | 78.1 |
| w/o MR | 77.3 | 77.3 | 77.1 | 77.2 |

## E    EVALUATION OF FORWARD TRANSFER AND BACKWARD TRANSFER

We compare the backward transfer and forward transfer performance of Q-tuning with the competitors using the metrics defined by (Lopez-Paz & Ranzato, 2017) in the long-sequence experiments. Figures 3, 4 and 5 show the forward transfer scores of the order 8 task sequence, Figures 6, 7 and 8 show the forward transfer scores of the order 9 task sequence, and Figures 9, 10 and 11 show the forward transfer scores of the order 10 task sequence.

Figures 12, 13 and 14 show the backward transfer scores of the order 8 task sequence, Figures 15, 16 and 17 show the backward transfer scores of the order 9 task sequence, and Figures 18, 19 and 20 show the backward transfer scores of the order 10 task sequence. In these backward transfer measurements, the score 0 stands for not forgetting old tasks.

We also report the evolution of the average accuracy over learning new tasks (Lopez-Paz & Ranzato, 2017) in Figure 21.

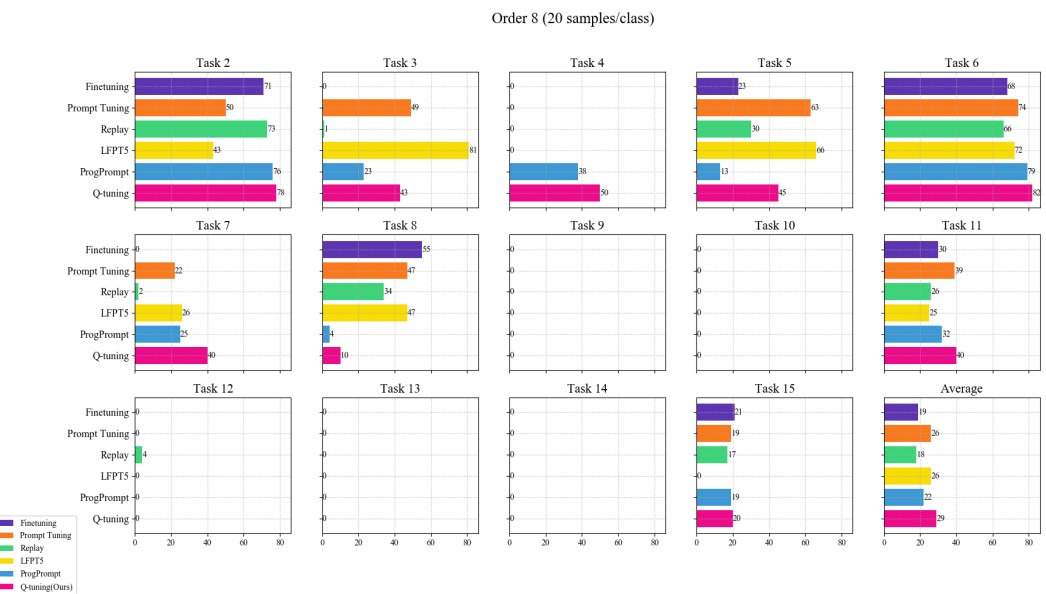

Figure 3: Forward transfer score of different approaches on the order 8 (20 samples/class).

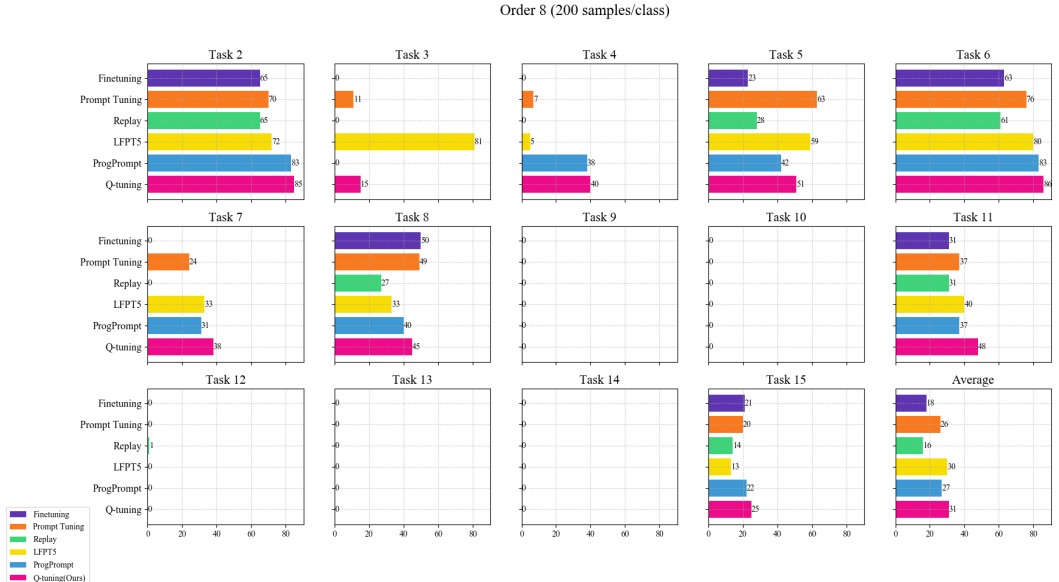

Figure 4: Forward transfer score of different approaches on the order 8 (200 samples/class).

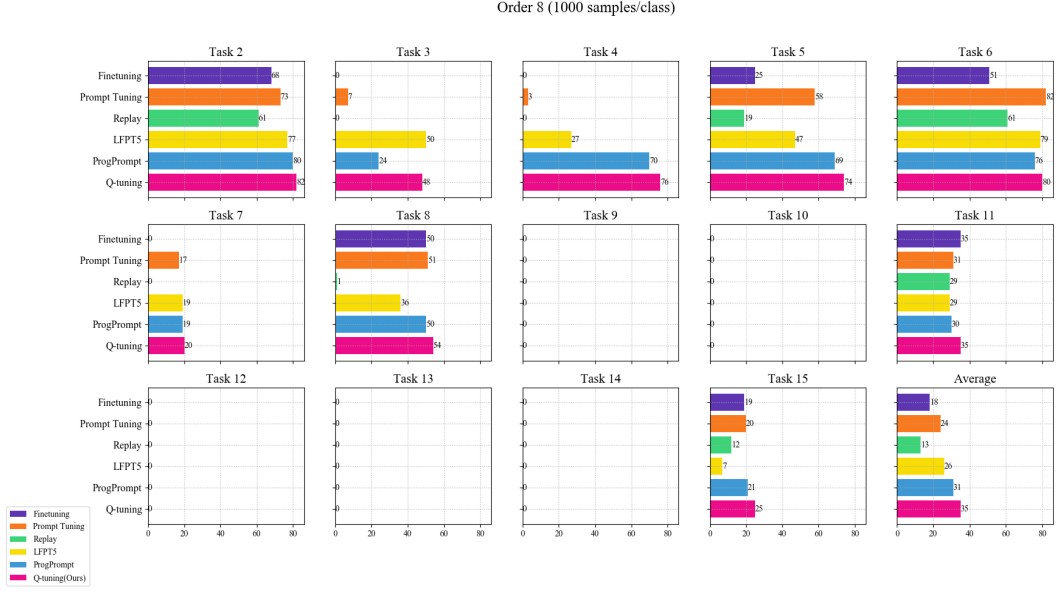

Figure 5: Forward transfer score of different approaches on the order 8 (1000 samples/class).

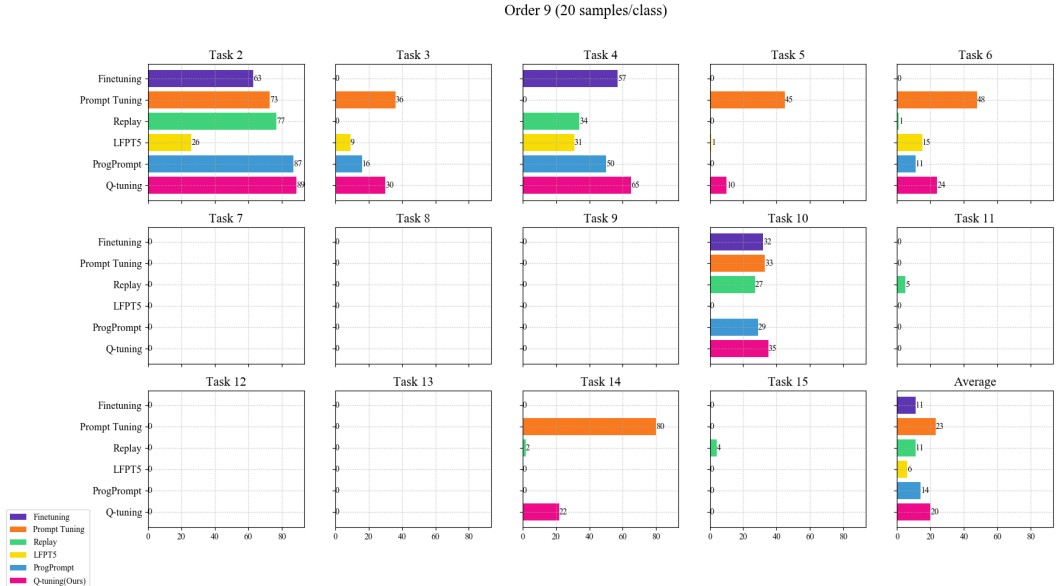

Figure 6: Forward transfer score of different approaches on the order 9 (20 samples/class).

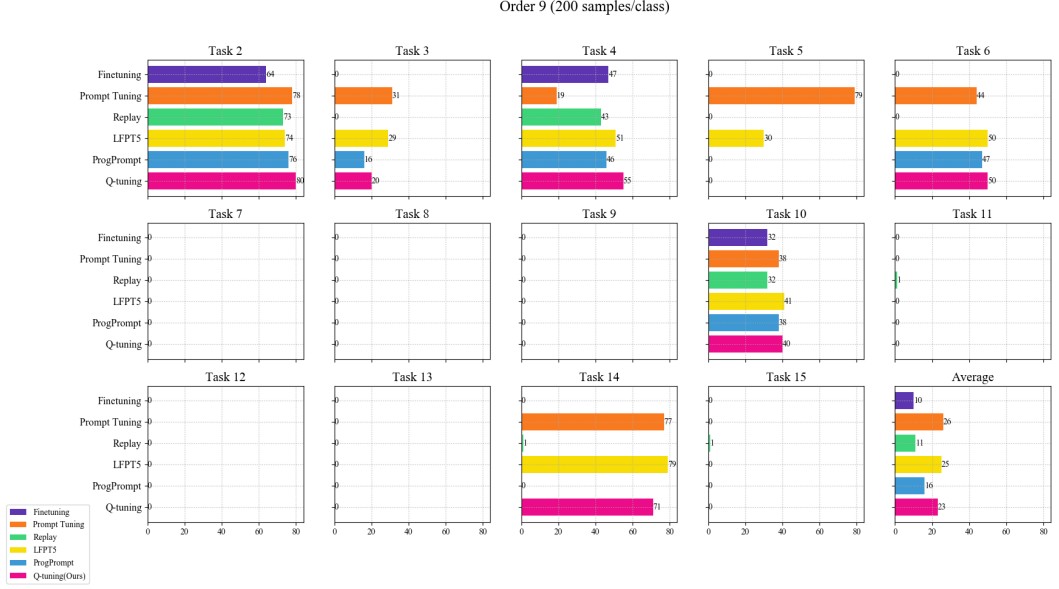

Figure 7: Forward transfer score of different approaches on the order 9 (200 samples/class).

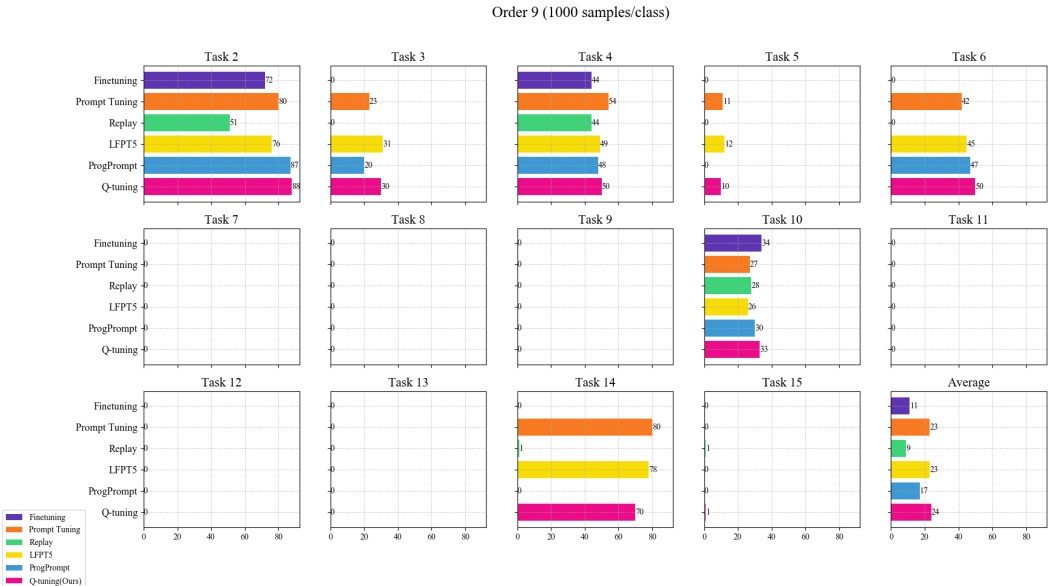

Figure 8: Forward transfer score of different approaches on the order 9 (1000 samples/class).

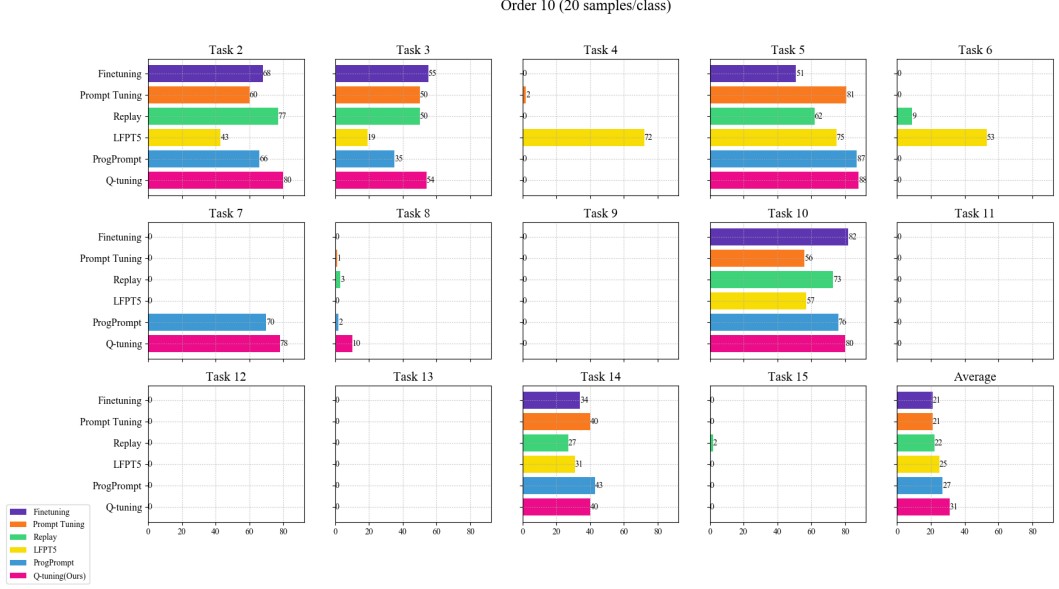

Figure 9: Forward transfer score of different approaches on the order 10 (20 samples/class).

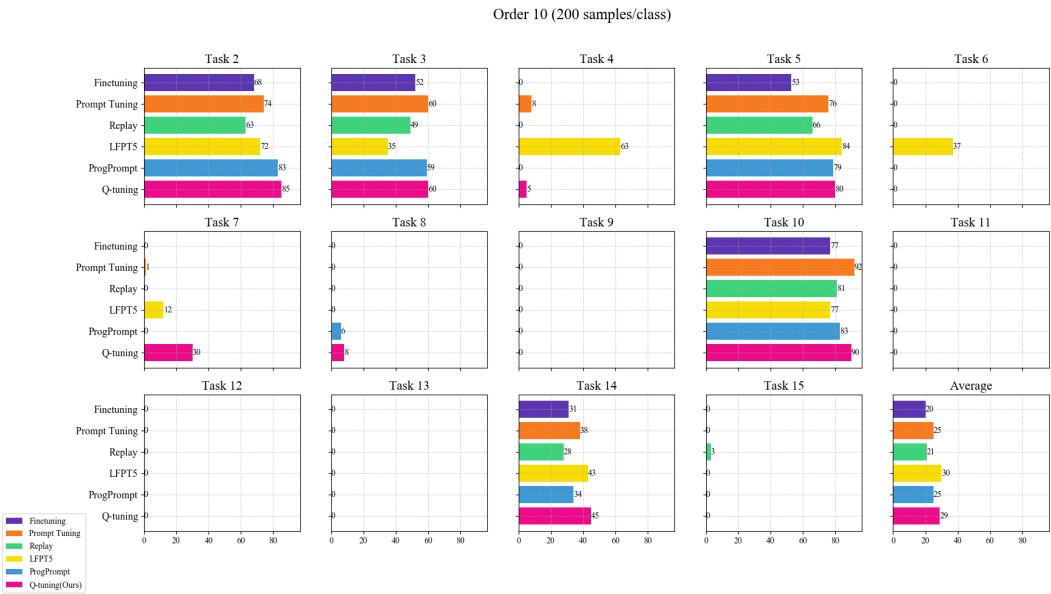

Figure 10: Forward transfer score of different approaches on the order 10 (200 samples/class).

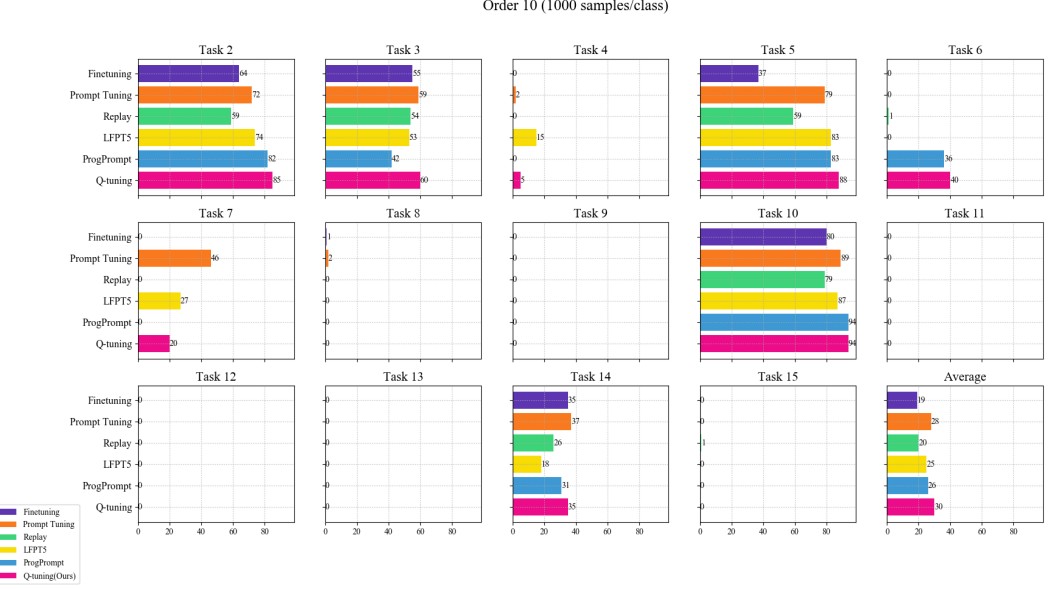

Figure 11: Forward transfer score of different approaches on the order 10 (1000 samples/class).

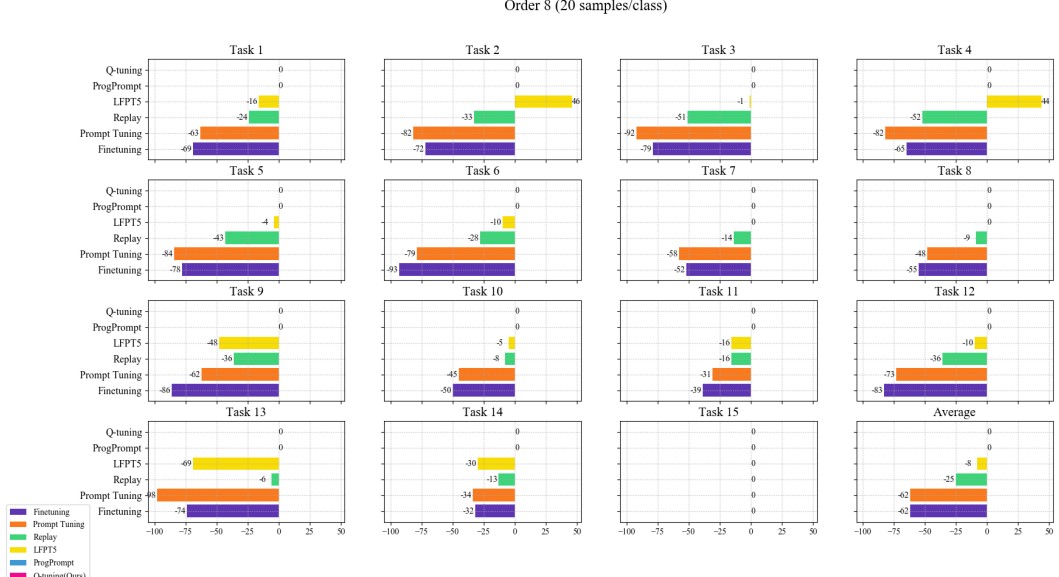

Figure 12: Backward transfer score of different approaches on the order 8 (20 samples/class).

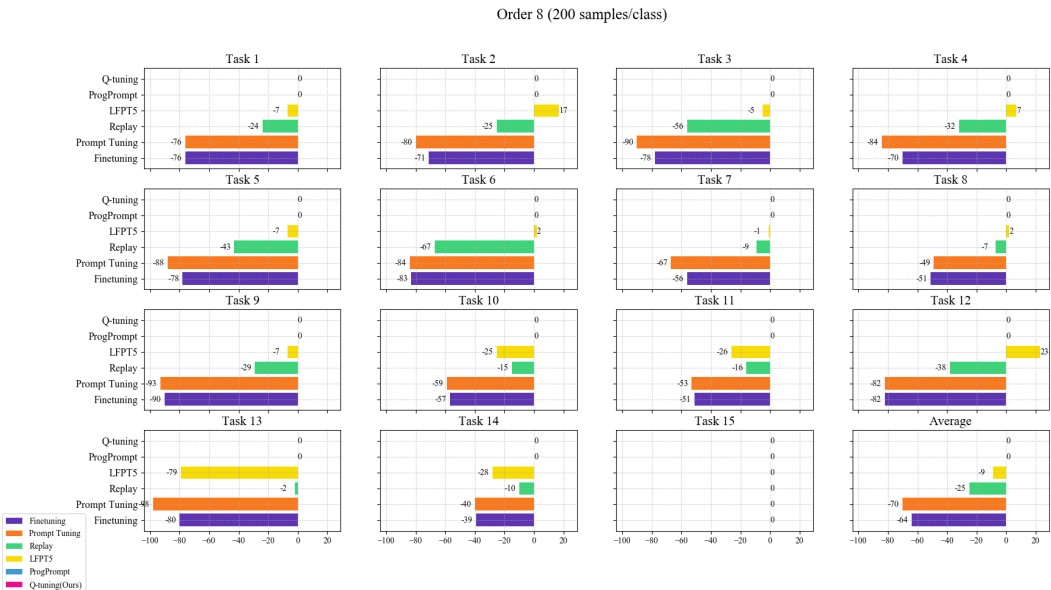

Figure 13: Backward transfer score of different approaches on the order 8 (200 samples/class).

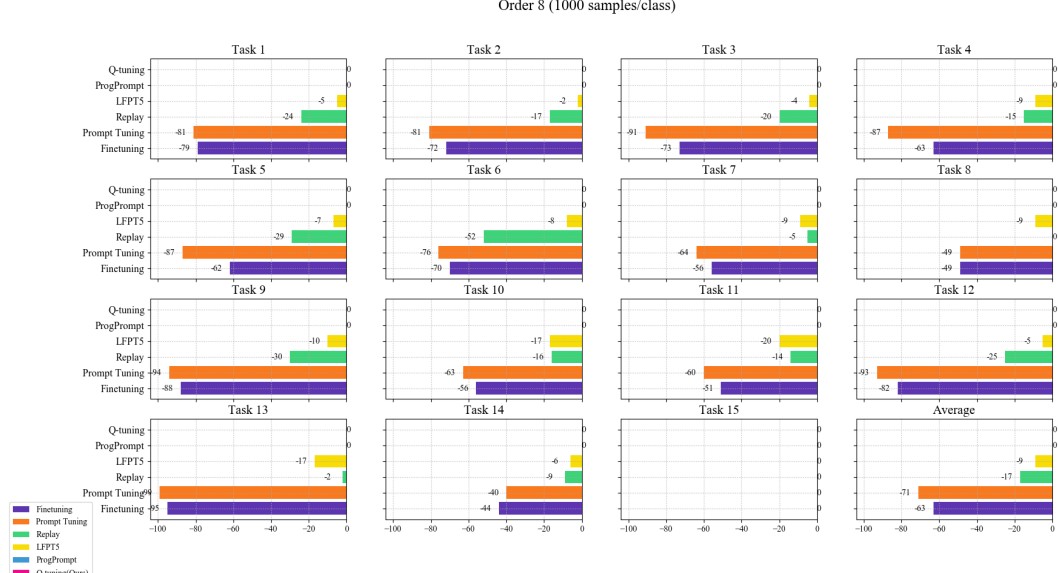

Figure 14: Backward transfer score of different approaches on the order 8 (1000 samples/class).

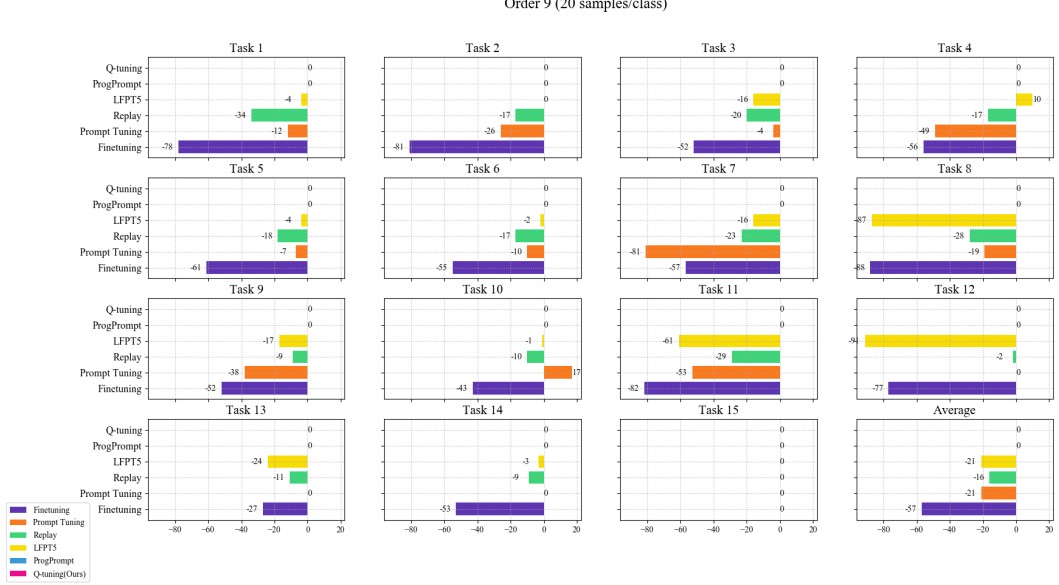

Figure 15: Backward transfer score of different approaches on the order 9 (20 samples/class).

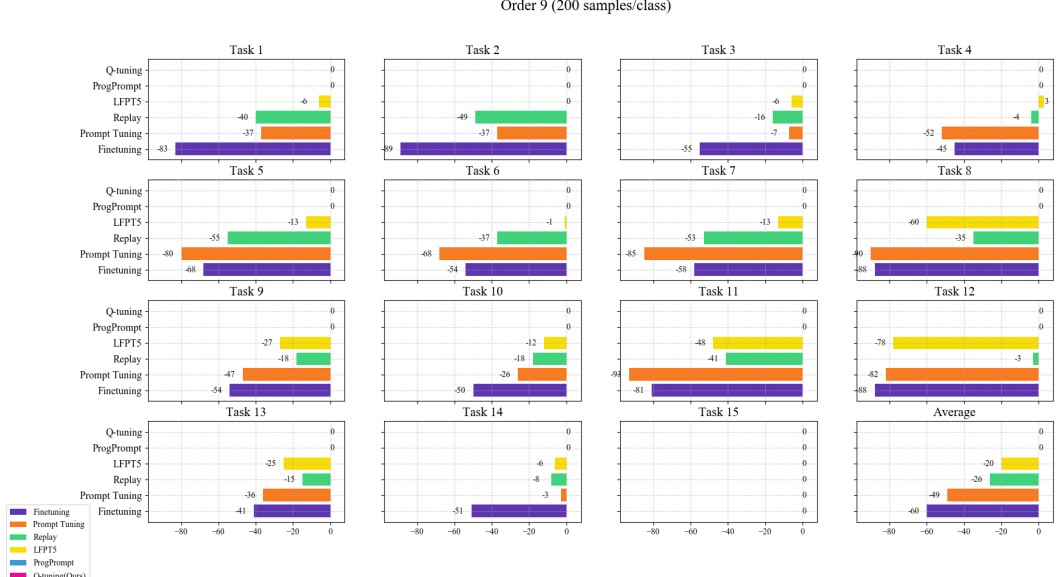

Figure 16: Backward transfer score of different approaches on the order 9 (200 samples/class).

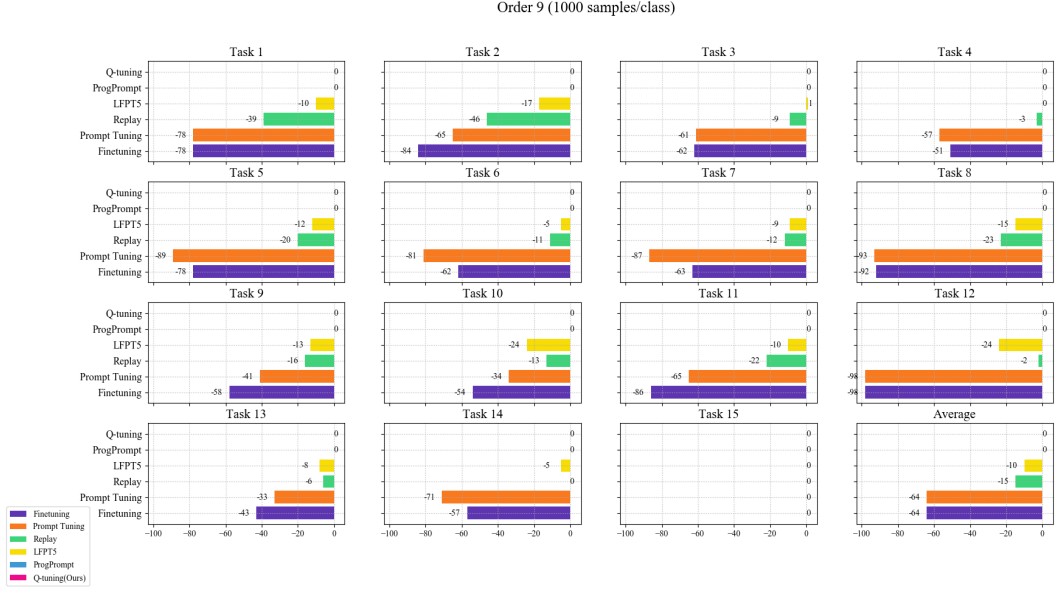

Figure 17: Backward transfer score of different approaches on the order 9 (1000 samples/class).

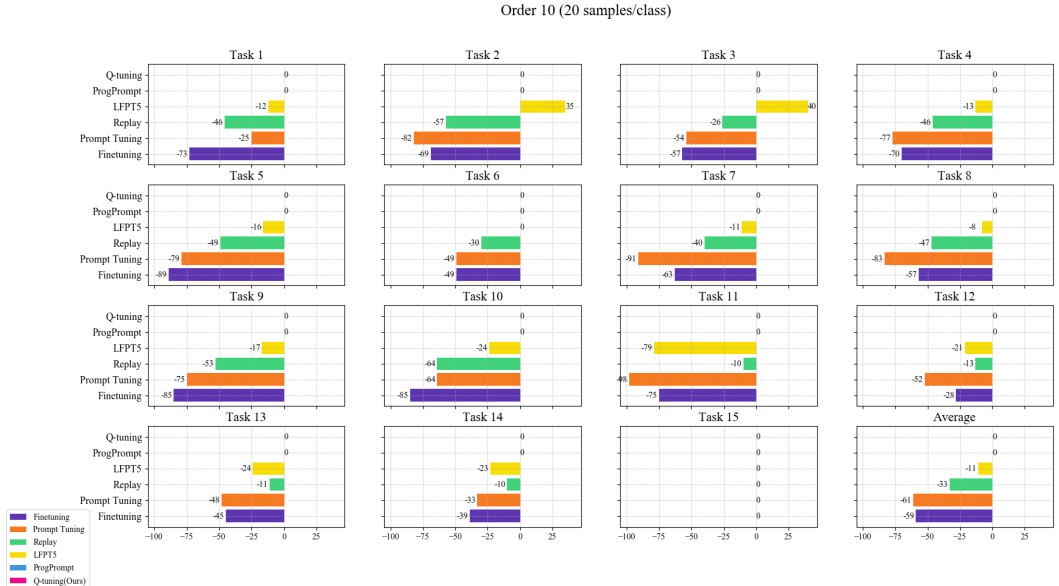

Figure 18: Backward transfer score of different approaches on the order 10 (20 samples/class).

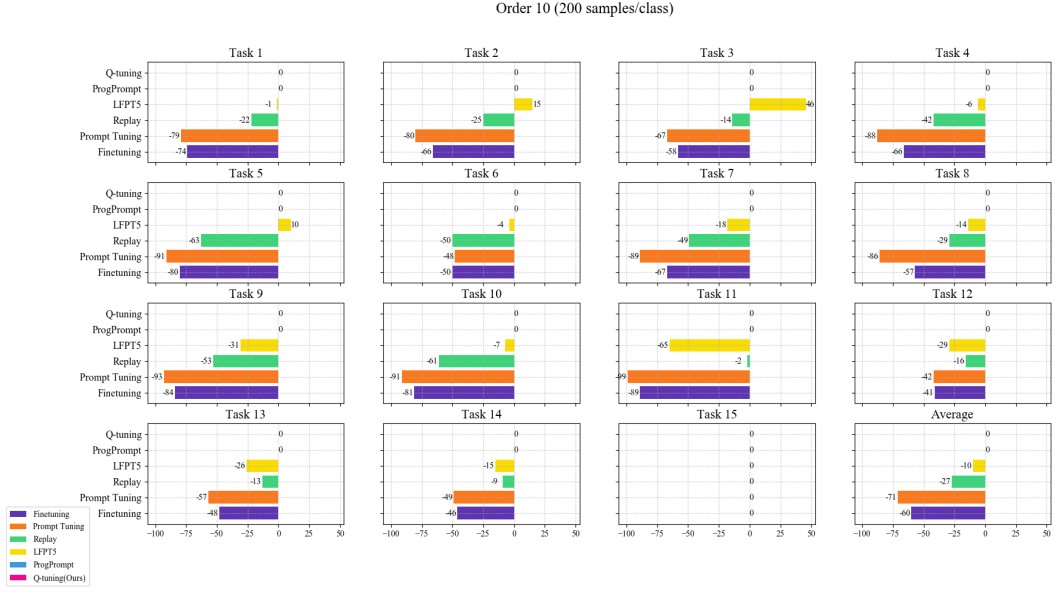

Figure 19: Backward transfer score of different approaches on the order 10 (200 samples/class).

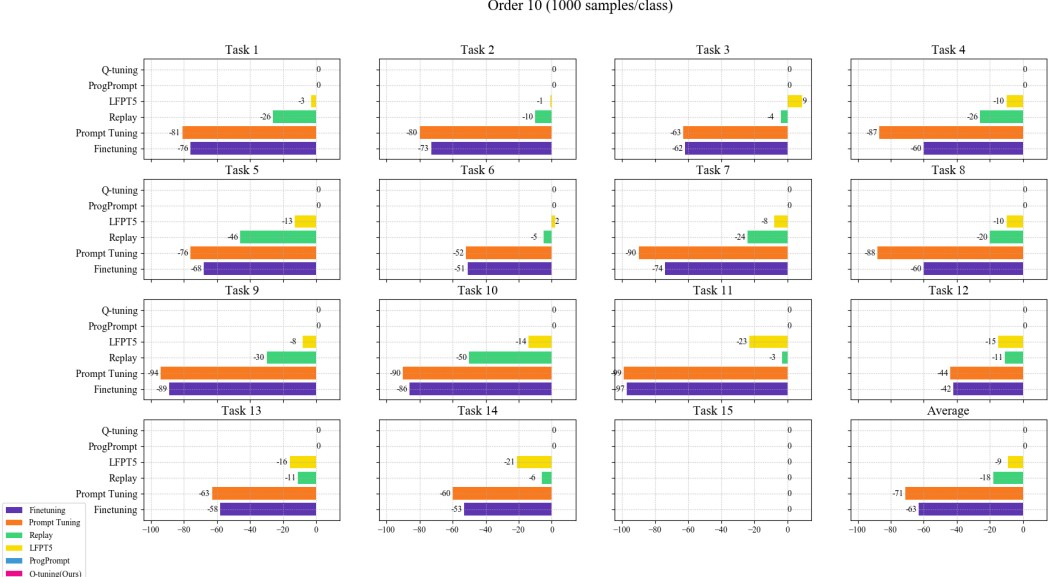

Figure 20: Backward transfer score of different approaches on the order 10 (1000 samples/class).

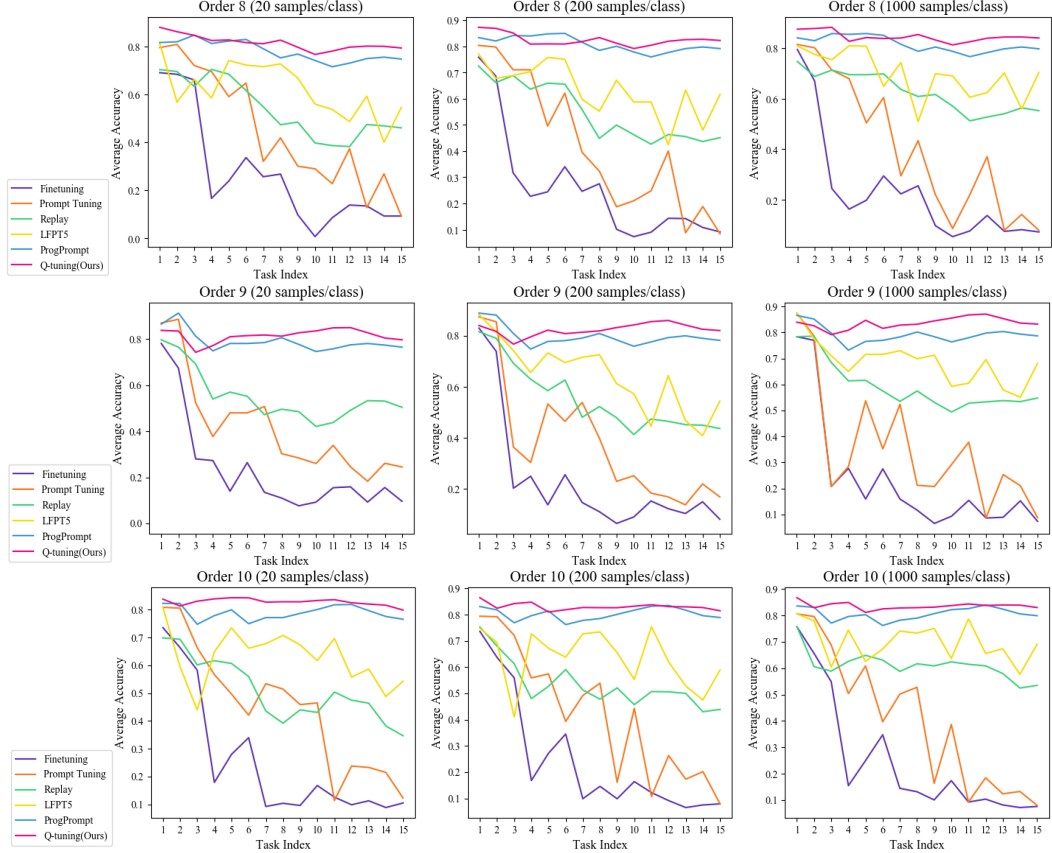

Figure 21: Evolution of average accuracy after learning new tasks.

