# OpenReview forum: "Q-Tuning: Continual Queue-based Prompt Tuning for Language Models"
_ICLR.cc/2024/Conference — ICLR 2024 Conference Withdrawn Submission_

### Official Review · Reviewer_r9Qa · 2023-10-31

**Soundness:** 3 good
**Presentation:** 4 excellent
**Contribution:** 2 fair
**Rating:** 5
**Confidence:** 5

**Summary:**

This paper proposes Q-tuning, an efficient continual prompt learning for large language models. Prior works in continual prompt learning suffered from an increasing size of prompt as the task number increased. To address this challenge, the authors proposed a way to have a fixed-size prompt by removing the prompts that have non-relevant information to the next task (i.e., de-Q). Also, to better learn the prompts (for forward transfer), the author proposes Q-Prompt Ensemble, which selects the most relevant prompt of the prior tasks when learning a new task by using a low-rank matrix (rank of one). Finally, the author proposes a globally shared prefix prompt for information retention by maximizing the mutual information of the prediction when using the previous task prompts (not pruned yet by the de-Q).

**Strengths:**

(1) The overall method is sound and effective. Tackled the existing limitation of continual prompt tuning, i.e., the number of prompt increases as the task increase, with a reasonable approach.

(2) The overall writing is very clear, and the presentation is well organized.

(3) The experiments are extensive. Also, the proposed method shows quite good overall results. More analysis by plotting the accuracy and reporting backward transfer will improve the paper.

**Weaknesses:**

(1) Insufficient related work. The tackled problem (i.e., progressively extending prompts) is a general limitation of expanding networks in continual learning literature [1,2]. There already exist papers that tackle such problems in continual learning [3]; hence, the authors should more rigorously do the literature review.

(2) Important baseline is missing [3]. For instance, Progress \& compress [3] tackles the same problem, i.e., tackles the problem of the progressively expanding network (although they did not use it for LLMs; since it was quite old paper). The overall concept is highly similar, for instance, using past knowledge for better forward transfer and compressing the knowledge into a compact representation. Since, [3] can be used for any model (i.e., model agnostic) and can be easily extended to continual prompting (i.e., progressively expanding prompts), the authors should compare the results. Also, the authors should rigorously discuss the similarities and dissimilarities.

(3) It is quite hard to understand that Q-Tuning outperforms ProgPrompt (or Full prompts, MTL) under a long sequence of tasks. Since other methods have a bigger number of parameters (as the prompt size increases), I believe the performance also should be better. Can the author give an explanation or intuition behind these results? Although I can understand that Q-Tuning can have a better forward transfer than ProgPrompt (due to the Ensemble), it is a little awkward that Q-Tuning with a limited capacity shows better results than Full prompts of Q-Tuning.

(4) De-Q rule is indeed reasonable for better forward transfer (as it retains the most relevant prompt for the new task), but it may have a negative effect on the backward transfer (i.e., the forgetting).

(5) The paper only reports the overall performance in the main table, but dividing the results into i) backward, ii) forward, and iii) overall will be more informative.

(6) Somewhat hard to understand the intention of $\theta_{P}^{*}$, i.e., the prefix prompt for global knowledge sharing.
- i) First, even though a single prefix prompt well preserves the knowledge by maximizing Eq (5), the optimized prefix will be concatenated with the pruned prompt (i.e., $[\theta_{P}^{*}, Q^{-1}]$). Therefore, the resulting prompt does not explicitly maximize the mutual information.
- ii) I think this paper mainly targets forward transfer (although I think backward transfer is also needed for measurement), but this component seems to be presented for backward transfer.
- I think analyzing the effect of $\theta_{P}^{*}$ is needed under both metrics, i.e., forward and backward transfer.

(7) (minor) The novelty is somewhat limited, e.g., the Q-prompt ensemble is already proposed by the prior work [4] (also, the term 'Ensemble' is quite confusing, as it does not use multiple models. Maybe not using a new terminology for methods that prior work has proposed will be better), and the concept is quite similar as [3] (and the authors did not compare or highlight the difference in the main text).

[1] Progressive Neural Networks, 2016\
[2] Lifelong Learning with Dynamically Expandable Networks, ICLR 2018\
[3] Progress & Compress: A scalable framework for continual learning, ICML 2018\
[4] Multitask Prompt Tuning Enables Parameter-efficient Transfer Learning, ICLR 2023

**Questions:**

Can the author report the plot of the individual task accuracy curve during continual learning (just for the main baseline, ProgPrompt, and Q-Tuning) ? Such a plot is very helpful in understanding the overall behavior during continual adaptation, e.g., backward, forward transfer.

See some examples below\
Figure 3 of Online Meta-Learning, ICML 2019\
Figure 2 of Online Fast Adaptation and Knowledge Accumulation: a New Approach to Continual Learning, NeurIPS 2020\

---

Also, is the size of GPT-4 known to the community (just out of curiosity)? Will be great to include the reference in Line 22.

---

> ### Author Response · Authors · 2023-11-15
> **To Reviewer r9Qa**
>
> Thank you very much for your constructive and informative review and detailed comments. We are encouraged to know that you found our paper well-written, our proposed method sound and effective and our experiments extensive. We have updated the paper with additional metrics and plots, and we also added some useful references in the related work section based on your recommendation. Below are our answers to all your questions:
>
> ---
> **Q1: Insufficient related work [1,2,3].**
>
> **A1**: Thank you for your valuable feedback. We have added more related works in the revised version (Lines 77-82). We do not include them in our comparison because:
> 1. The method [1] needs to maintain previous networks to train the network for a new task, making it not suitable for current large language models. We aim to enable lifelong learning with a single network, which is why we only surveyed papers in that domain.
> 2. The method [2] dynamically decides which sub-network to train for a sequence of tasks. However, our scope is in the prompt tuning of large language models, where keeping the model frozen is necessary given the size of recent language models. In such cases, even training a sub-network within the language models will become unmanageable. Moreover, [2] is designed based on multi-layer neural networks such as DNN, and its efficacy in transform-like architecture has yet to be tested.
> 3. Our scope is in prompt tuning. This is because given the large size of modern language models, we'd like to avoid any approaches that involve the training of the whole network or subnetwork. Another reason is, that prompt tuning is naturally immune to catastrophic forgetting (CF), because we can cheaply store task-specific prompts, and retrieve the related prompts for inference when dealing with the previous task. In comparison, the work [3] leverages distillation and knowledge preservation (i.e., EWC). Although it can mitigate CF while not completely eliminating it as prompt tuning does.
>
>
> ---
> **Q2: Important baseline is missing [3]. For instance, Progress & compress [3] tackles the same problem ... since it was quite old paper). The overall concept is highly similar, for instance, using past knowledge for better forward transfer and compressing the knowledge into a compact representation. Since, [3] can be used for any model (i.e., model agnostic) and can be easily extended to continual prompting (i.e., progressively expanding prompts), the authors should compare the results. Also, the authors should rigorously discuss the similarities and dissimilarities.**
>
> **A2**:
> Here, we discuss the similarities and dissimilarities between Q-tuning and the paper [3] from the perspectives of the method paradigm and the application area.
>
> * Similarities: As far as we can understand, Q-tuning is partially conceptually similar to the paper [3] at the framework level, where they both adopt a shared knowledge base and a task-specific knowledge base.
>
> * Dissimilarities:
>
>   **(1)** The proposed Q-tuning following the prompt tuning paradigm is naturally immune to catastrophic forgetting (CF) because we can cheaply store task-specific prompts, and retrieve the related prompts for inference when dealing with the previous task. But the paper [3] can only mitigate the CF problem while not eliminating it;
>
>   **(2)** Our scope is in the prompt tuning area. The modern language models often than not involve billions of parameters. It is highly expensive and not smart to tune the whole parameters or subnetworks of the pretrained large-scale backbones for each new task. Prompt tuning paves the way to efficiently tuning large-scale models.
>
>   **(3)** Based on the prompt tuning technique, the proposed Q-tuning focuses on solving the continual prompt tuning problem.  Specifically, continual prompt tuning solves the learning problem defined by Eq. (2) as introduced in Lines 96~116. However, when the prompt list grows asymptotically (i.e. the model is set as a lifelong learner), training the extremely long prompt list becomes intractable due to the finite system resources. The proposed Q-tuning can reduce the computation cost regarding GPU memory and inference latency **from $\mathcal{O}(N^2)$ to $\mathcal{O}(1)$ for life-long learning scenarios**.
>
>   **(4)** Moreover, we propose a Memory Retention technique to reduce information loss when trimming the Q-prompt for lifelong learning.
>
> In summary, our Q-tuning is partially conceptually similar to the paper [3]. But there are many differences between them from the perspectives of the technique roadmap and the applications. Last but not least, we cannot find a public codebase for the paper [3]. If there is an accessible codebase, we are happy to compare it with their method.

---

> ### Author Response · Authors · 2023-11-15
> **(Continued) To Reviewer r9Qa**
>
> **Q3: It is quite hard to understand that Q-Tuning outperforms ProgPrompt (or Full prompts, MTL) under a long sequence of tasks. Since other methods have a bigger number of parameters (as the prompt size increases), I believe the performance also should be better. Can the author give an explanation or intuition behind these results?**
>
> **A3**: Compared with other approaches with a larger number of parameters, the performance gain of Q-Tuning mainly comes from:
> 1. The Q-prompt Ensemble module adaptively organizes previous prompts based on their relevance to the current task, instead of directly leveraging previous prompts as ProgPrompt did;
> 2. Although Q-tuning throws away previous prompts in a long task sequence, the DQ-PCA ensures that only the least relevant parts are discarded. In other words, the DQ-PCA can **remove some redundant information in previously trained  prompts**, hence improving the performance to some extent
> 3. The shared prompt also bring gains to the overall performance by facilitating the FWT.
>
> We have reported the ablation study results in Tables 5&6. We copy the results here.
>
> | Method (Short Sequence) | 16 samples | 200 samples | 1000 samples| Average
> | -------- | -------- | -------- | -------- | -------- |
> | ProgPrompt     | 74.5 | 79.8 | 79.8| 78.0 |
> | + Ensemble   | 75.2 | 80.9 | 80.4 | 78.8  |
> | + Shared Prompt   | 75.1| 80.6| 80.9| 78.9  |
> | + Ensemble & Shared Prompt   |** 76.2** | **81.2**| **81.9**|** 79.7 (+1.7)**|
>
> | Method (Long Sequence) | 20 samples | 200 samples | 1000 samples| Average
> | -------- | -------- | -------- | -------- | -------- |
> | ProgPrompt     | 76.7| 80.8| 80.8| 79.4 |
> | + Ensemble   | 77.2| 81.1| 82.1| 80.2  |
> | + Shared Prompt   | 77.4| 81.1| 82.3| 80.3 |
> | + Ensemble & Shared Prompt   | **78.9** |**81.9**|**83.3**| **81.4 (+2.0)**|
>
> | Method (Extremely Long Sequence) | Order11 | Order12 | Order13 | Average |
> | -------- | -------- | -------- | -------- |-------- |
> | ProgPrompt (w/o Q-prompt) | Out of Memory | Out of Memory | Out of Memory | -|
> | Q-prompt (appending task-specific prompts)     | 86.8 | 87.3 | 87.7 |  87.3|
> | Q-prompt + shared prompt  | 87.6 | 88.1 | 88.7 | 88.1 |
> |Q-prompt + shared prompt +Ensemble   | 89.8 | 89.4 | 90.1 | 89.8 |
>  |Q-prompt + shared prompt +Ensemble + MR    | **90.9**| **90.6**| **90.8** | **90.8**|
>
> We can observe that the ensemble, shared prompt and MR loss contribute to the improvement, especially for the (extremely) long sequence experiments.
>
> ---
> **Q4:Although I can understand that Q-Tuning can have a better forward transfer than ProgPrompt (due to the Ensemble), it is a little awkward that Q-Tuning with a limited capacity shows better results than Full prompts of Q-tuning.**
>
> **A4**: As for the comparison with the full prompt, we can see the performance is almost identical on average, as in Table 2. We believe the slightly better average performance of DQ-PCA is purely because of numerical fluctuation. E.g, full prompt outperforms DQ-PCA in some columns.  In Table 4, we can observe that the Full prompts of Q-tuning outperform the Q-prompt by setting ${Q}_{size}$ to 10 and 5. Indeed, the performance of the trimmed Q-prompt matches that of the full Q-prompt, showcasing the effectiveness of the elaborate De-Q algorithm in preventing information loss when trimming the Q-prompt.
>
> ---
>
> **Q5: De-Q rule is indeed reasonable for better forward transfer (as it retains the most relevant prompt for the new task), but it may have a negative effect on the backward transfer (i.e., the forgetting).**
>
> **A5**:
> 1. Q-tuing does not consider the backward transfer because all the previously trained prompts and the pretrained backbone are fixed. In this case, we cannot update previous statuses to enable the backward transfer. For evaluation, please refer to the added results in the following responses: Q6&A6.
> 2. Q-tuing solves the catastrophic forgetting problem.  This is because we can cheaply store the task-specific prompts and different snapshots of the shared prompts. In this way when we do inference for previous tasks, we can simply use the corresponding task-specific prompts and shared prompts.

---

> ### Author Response · Authors · 2023-11-15
> **(Continued) To Reviewer r9Qa**
>
> **Q6: The paper only reports the overall performance in the main table, but dividing the results into i) backward, ii) forward, and iii) overall will be more informative.**
>
> **A6**: Thanks for the suggestions! Following ProgPrompt [5], we provide additional experiments on the long-sequence benchmarks: forward transfer (FWT), backward transfer (BWT), and average accuracy evolution, as defined by Lopez-Paz et al.[6].
>
> Here is a short summary of our results on 15-task experiments with the T5 model:
>
> 1. **Forward transfer** (higher is better)
> The table below shows the average FWT across the 3 task orders (Order 8, Order 9, Order 10) of long-sequence experiments:
>
>   | Method | Few-shot (20/class) | Full-shot |
>   | -------- | -------- | -------- |
>   | Finetune | 16.9    | 16.0    |
>   | Replay   | 16.8	  | 14.0   |
>   | Prompt Tuning   | 23.1 |	24.9   |
>   | Per-task Prompts | 0 | 0 |
>   | LFPT5 | 18.9 | 24.9 |
>   | ProgPrompts | 21.1 | 24.7 |
>   | Q-tuning (Ours) | **26.7** | **29.7** |
>
> The full FWT results on all tasks across the three orders in few-shot settings (20, 200, 1000 samples per class) are available in Appendix E (Figures 3-11) and the anonymous link:   https://drive.google.com/drive/folders/1xtch_kyckijOVBzzQQA6RRO8XyktIDsy?usp=sharing
>
>
> 2. **Backward transfer** (higher is better, 0 = no forgetting)
> The table below shows the average BWT across the 3 task orders (Order 8, Order 9, Order 10) of long-sequence experiments:
>
> | Method | Few-shot (20/class) | Full-shot |
> | -------- | -------- | -------- |
> | Finetune | -59.5    | -63.5    |
> | Replay   | -24.7	  | -18.0    |
> | Prompt Tuning   | -47.9 |	-71.0   |
> | Per-task Prompts | **0** | **0** |
> | LFPT5 | -13.5 | -8.8 |
> | ProgPrompts | **0** | **0** |
> | Q-tuning (Ours) | **0** | **0** |
>
> The full BWT results on all tasks across the three orders in few-shot settings (20, 200, 1000 samples per class) are available in Appendix E (Figures 12-20) and the anonymous link:
> https://drive.google.com/drive/folders/1RE0wvRBk9BhRf7MGEnwt1p7hEQScpcPA?usp=sharing
>
>
> 3. **Average accuracy**
>  Evolution of average accuracies is shown in Appendix E (Figure 21) and the anonymous link: https://drive.google.com/file/d/1ZJYrb83uCKuTJsa_JK8xV1KjniOKI6OQ/view?usp=sharing
>
> All the above results are added to the Appendix E of the updated paper.
>
> **Summary of the Results:**
> We observe that the proposed Q-tuning has the best BWT score and the best FWT score in both few-shot and full-shot continual learning setups. Notablely, Q-tuning, following the prompt tuning paradigm, is able to eliminate the catastrophic forgetting problem, because for a new task, all previously tuned prompts are frozen. Moreover, Q-tuning is able to achieve life-long learning ability while achieving FWT in a limited computation resource. These two advantages of Q-tuning make it outperform the baseline [3] mentioned by the reviewer.
>
>
> ---
>
> **Q7: Even though a single prefix prompt well preserves the knowledge by maximizing Eq (5), the optimized prefix will be concatenated with the pruned prompt (i.e., $[\theta^{i-1}_{\mathcal{P}^{\ast}},\mathcal{Q}^{i-1}]$). Therefore, the resulting prompt does not explicitly maximize the mutual information.**
>
> **A7:** For the continual learning task, we assume the knowledge transfer satisfies the Markov process, i.e. the current status of the prompts is only determined by their last status. Hence, in Eq.(5), we maxmize the mutual information between the **current shared prompt  $\theta^{i}_{\mathcal{P}^{\ast}}$ of the i-th status and the prevous i-1 th status.** However, in practice, the ideal FWT does not exist, and the transferred knowledge to the shared prompt $\theta^{i}_{\mathcal{P}^{\ast}}$ will decay along with the coming of new tasks, resulting in the loss of mutual information. This phenomenon aligns with the well-known brain memory management. When we learn new things, the memory will decay over time.

---

> > ### Author Response · Authors · 2023-11-15
> > **(Continued) To Reviewer r9Qa**
> >
> > **Q8: The intention of $\theta_{\mathcal{P}^{\ast}}$. "I think this paper mainly targets forward transfer (although I think backward transfer is also needed for measurement), but this component seems to be presented for backward transfer."**
> >
> > **A8**: The intention of the shared prompt $\theta_{\mathcal{P}^{\ast}}$ is to avoid the cumulated information loss when we trim the Q-prompt.
> >
> > As introduced in Lines 100~116 and the Sec3.2 of the paper, the prompt list, i.e. Q-prompt will become intractable due to the finite system resources when the number of observed tasks grows asymptotically (i.e. the model is set as a lifelong learner). In this context, the training and inference complexity of maintaining the prompt list scales as $\mathcal{O}(N^2)$ for transformer based models regarding the number of tasks.  Hence, we propose the De-Q algorithm to trim the Q-prompt in order to maintain a controllable cost for life-long learning.  However, trimming the Q-prompt will cause information loss. This motivates us to add a globally shared prompt $\theta_{\mathcal{P}^{\ast}}$ to retain this lost information by using the novel Memory Retention regularization defined by Eq.(5). Because the shared prompt is used to solve the information loss problem for life-long learning while enhancing the forward transfer, we have conducted the ablation studies regarding the shared prompt on extremely long task sequences (70 different tasks). The results are reported in Table 6.
> >
> > ---
> >
> > **Q9: Minor concern on the novelty of ensemble**
> >
> > **A9**: We agree that the ensemble model itself is not novel. The ensemble strategy is inspired by the paper [4], which has been cited in Lines 135 - 136. Our core contribution is the design of Q-Tuning to enable lifelong learning of extremely long task sequences, and the Q-prompt Ensemble is one of the many feasible approaches we can think of to adaptively incorporate previous prompts. We did not thoroughly evaluate different ensemble models given that it's just a feature but not the focus of our approach.
> >
> > ---
> > **Q10: Plot of individual task accuracy curve.**
> >
> > **A10**: Thanks for the advice. We have added the evolution of average accuracies in the updated paper. The results are presented in Appendix E (Figure 21) and the anonymous link: https://drive.google.com/file/d/1ZJYrb83uCKuTJsa_JK8xV1KjniOKI6OQ/view?usp=sharing
> >
> > ---
> > **Q11: Number of parameters in GPT-4**
> >
> > **A11**: GPT-4's size (1.76 trillion) is known to the community according to multiple sources such as https://medium.com/@mlubbad/the-ultimate-guide-to-gpt-4-parameters-everything-you-need-to-know-about-nlps-game-changer-109b8767855a and https://the-decoder.com/gpt-4-has-a-trillion-parameters/#:~:text=Further%20details%20on%20GPT%2D4's,Mixture%20of%20Experts%20(MoE). We have added the refence in the resubmitted version.
> >
> > ---
> > [1] Progressive Neural Networks, 2016
> >
> > [2] Lifelong Learning with Dynamically Expandable Networks, ICLR 2018
> >
> > [3] Progress & Compress: A scalable framework for continual learning, ICML 2018
> >
> > [4] Multitask Prompt Tuning Enables Parameter-efficient Transfer Learning, ICLR 2023
> >
> > [5] Progressive Prompts: Continual Learning for Language Models, ICLR 2023
> >
> > [6] Gradient episodic memory for continual learning, NeurIPS 2017

---

> > > ### Author Response · Authors · 2023-11-19
> > > **Followup Response to Reviewer r9Qa**
> > >
> > > Dear Reviewer r9Qa,
> > >
> > > We sincerely thank you again for reviewing our paper, and we appreciate your insightful advice regarding adding divided results to support our proposed technique. In our response, we have endeavored to elucidate the difference between our method and the baselines you referenced. Furthermore, we have provided a comprehensive clarification of our motivation and addressed the concerns about our empirical results. Should you have any further questions, please do not hesitate to reach out to us; we would be delighted to assist.
> > >
> > > Best,
> > >
> > > Paper 383 Authors

---

### Official Review · Reviewer_tNki · 2023-11-01

**Soundness:** 3 good
**Presentation:** 3 good
**Contribution:** 3 good
**Rating:** 5
**Confidence:** 5

**Summary:**

The paper proposes a new prompt-based continual learning method for sequential language tasks. The core components are threefold: globally shared prefix prompt, MLP (two-layered and residual) parameterized prompts, and a De-q process to keep the free capacity of prompt memory for future tasks. The model updates prefix prompts through ensemble updates of past-task prompts and temporally learnable weights to retain past-task information and prevent overfitting to current tasks. For De-q, when the buffer capacity reaches the pre-defined maximum size,  the model decomposes learned queries via the SVD-based technique. The proposed method is extensively validated using Bert/T-5 models on multiple task sequences and achieved improved performance.

**Strengths:**

The paper addresses two challenges in prompt-based continual learning - preventing forgetting efficiently and keeping the prompt capacity reasonable even on training long task sequences. The paper is basically well-described and provides extensive evaluations on multiple benchmark task sequences using Bert/T-5 backbones. Also, unlike Progprompt, a prompt-based continual learning baseline, the suggested method can handle long task sequences (~70 tasks) without suffering from memory overhead issues.

**Weaknesses:**

- Lack of comparison with prompt-based continual learning methods. Only Progprompt can be a straightforward baseline in view of 'prompt-based continual learning', which is insufficient since there are multiple strong prompt-based continual learning approaches [1,2,3].

- Insufficient forward knowledge transfer evaluation. The paper only compares the FKT with a simple prompt-tuning. Throughout the paper, the authors emphasize multiple times, but, this is not that helpful to validate the strength of the proposed method on FKT. I strongly recommend evaluating it with more diverse continual learning methods, including [4,5], which also empirically emphasize forward transfer improvements in language continual learning.

- Missing ablation studies like efficiency. Recently, some continual learning researchers claim that the training time/real-time computational costs matter in real continual learning scenarios rather than memory occupancy, as the monetary cost for memory (e.g., HDD) is much cheaper than computations (e.g., GPUS) [6]. So, I also recommend providing the wall-clock evaluation/ trainable parameter counts compared to recent prompt-based continual learning methods that I introduced.



[1] James Seale Smith, et al., Coda-prompt: Continual decomposed attention-based prompting for rehearsal-free continual learning. CVPR 2023.
[2] Yabin Wang, et al., S-prompts learning with pre-trained transformers: An occam’s razor for domain incremental learning. NeurIPS 2022.
[3] Zifeng Wang, et al., Dualprompt: Complementary prompting for rehearsal-free continual learning. ECCV 2022.
[4] Wenpeng Yin, et al., ConTinTin: Continual Learning from Task Instructions, ACL 2022.
[5] Qi Zhu, et al., Continual Prompt Tuning for Dialog State Tracking, ACL 2022.
[6] Ameya Prabhu, et al., Computationally Budgeted Continual Learning: What Does Matter? CVPR 2023.

**Questions:**

.

---

> ### Author Response · Authors · 2023-11-15
> **To Reviewer tNki**
>
> Thank you very much for your constructive review and feedback. We are pleased and encouraged to learn that you found our idea promising, our proposed method novel and effective, and our paper well-written and validated. We also thank you for providing all the useful references, as we have added some of them in our related work section. We are addressing each of your comments below:
>
> ---
>
> **Q1**: **Lack of comparison with prompt-based continual learning methods. Only Progprompt can be a straightforward baseline in view of 'prompt-based continual learning', which is insufficient since there are multiple strong prompt-based continual learning approaches [1,2,3].**
>
> **A1**: Thank you for introducing these relevant papers. We didn't include them initially, as we mainly surveyed NLP-related works. We've added them to the resubmitted paper (Lines 79-80, 86-88 and 96-98). We'd like to elaborate on the difference between Q-tuning in the following aspects:
>
> 1. The work [1] uses an expanding prompt component set similar to the ProgPromt, which cannot solve an infinite number of tasks. As the new task number becomes larger, there will be a point when the training fails due to out of memory. This limitation actually motivates the design of the Q-prompt and De-Q algorithm of Q-tuning to enable life-long learning.
> 2. The paper [2] cumulates a prompt pool similar to ProgPrompt, which is considered a baseline approach in Q-tuning. Besides, it cannot tackle extremely long task sequences. In addition, [2] is designed for CLIP-like architectures, whereas Q-tuning is agnostic to model backbones including encoder-only classification models (e.g., BERT) and generative models (e.g., T5 and other advanced language models).
> 3. A major contribution of DualPrompt [3] is to add the shared prompt and task-specific prompts to different layers of the backbone model. This layer-wise prompt configuration, however, necessitates the **prior determination of the total number of tasks**, which is orthogonal to our approach, as Q-tuning's focus is on enabling lifelong learning where **the total number of tasks is unknown and could be very large**.
>
> Based on the above points，the mentioned papers [1,2,3] should not be good baselines in our paper for language models and the experiment settings. Morever,  these papers focus on CV tasks and do not provide code for NLP experiments.
>
> ---
>
> **Q2: Evaluating the forward knowledge transfer with more diverse continual learning methods, including [4,5].**
>
> **A2**: We did not compare with the methods [4,5] because they are based on different settings. The paper [4] requires a few training tasks equipped with instructions and labelled examples to initialize a model, which is not available in our lifelong learning scenarios. The paper [5] leverages a data buffer from previously observed tasks to enable FKT, therefore it's not rehearsal-free. **Q-tuning focuses on the practical situation in lifelong learning where we do not have access to previous data and do not know the total number of incoming tasks**.
>
> ---
>
> **Q3**: **Missing ablation studies like efficiency. Recently, some continual learning researchers claim that the training time/real-time computational costs matter in real continual learning scenarios rather than memory occupancy, as the monetary cost for memory (e.g., HDD) is much cheaper than computations (e.g., GPUS) [6]. ....**
>
> **A3**: **The proposed Q-tuning aims to solve the computation costs in GPUs** rather than the HDD, **which can reduce the computation cost regarding GPU memory** from $\mathcal{O}(N^2)$ to $\mathcal{O}(1)$ for life-long learning scenarios. For the hardware storage overhead, the proposed Q-tuning assumes it is unlimited because it is very cheap.  The proposed Q-tuning maintains a prompt buffer, i.e. Q-prompt of a finite length rather than an infinite length, to make the computation cost for training and inference consistent when the number of tasks grows.
>
> We explain the motivation of Q-tuning in Lines 106-116, the introduction of the De-Q method in Lines 141-163, and the results in Table 3. In Table 3, the experiments demonstrate that the proposed Q-tuning is able to achieve continual prompt tuning across 70 different tasks, while the current SOTA ProgPrompt fails due to the GPU memory problem, i.e. the extremely high computation memory costs. Regarding the inference latency, the proposed Q-tuning can also reduce the computation cost from $\mathcal{O}(N^2)$ to a constant cost determined by the set Q-prompt size $\mathcal{Q}_{size}$, where $N$ stands for the number of tasks.

---

> > ### Author Response · Authors · 2023-11-15
> > **(Continued) To Reviewer tNki**
> >
> > Finally, we hope our clarifications can solve the reviewer's concerns. If there are more questions, please let us know. We are happy to discuss.
> >
> > ---
> >
> > [1] James Seale Smith, et al., Coda-prompt: Continual decomposed attention-based prompting for rehearsal-free continual learning. CVPR 2023.
> >
> > [2] Yabin Wang, et al., S-prompts learning with pre-trained transformers: An occam’s razor for domain incremental learning. NeurIPS 2022.
> >
> > [3] Zifeng Wang, et al., Dualprompt: Complementary prompting for rehearsal-free continual learning. ECCV 2022.
> >
> > [4] Wenpeng Yin, et al., ConTinTin: Continual Learning from Task Instructions, ACL 2022.
> >
> > [5] Qi Zhu, et al., Continual Prompt Tuning for Dialog State Tracking, ACL 2022.
> >
> > [6] Ameya Prabhu, et al., Computationally Budgeted Continual Learning: What Does Matter? CVPR 2023.
> >
> > [7] Xiang Lisa Li, et al., "Prefix-tuning: Optimizing continuous prompts for generation", ACL 2021.

---

> > > ### Author Response · Authors · 2023-11-19
> > > **Followup Response to Reviewer tNki**
> > >
> > > Dear Reviewer tNki:
> > >
> > > We would like to thank you again for the invaluable time you dedicated to reviewing our paper. In our response, we have clarified that the proposed Q-tuning focuses on the practical situation in lifelong learning where the total number of incoming tasks is unknown and the previous data is inaccessible. In addition, we have explained the motivation of Q-tuning, which is for reducing the computation cost regarding, e.g., GPU memory and inference latency from $\mathcal{O}(N^2)$ to $\mathcal{O}(1)$ for life-long learning scenarios rather than the hardware disk, e.g., HDD. More results of FWT, BWT, and the evolution of average accuracy are added to Appendix E to provide a more comprehensive comparison with previous methods. We hope that our response can address your concerns. Please feel free to share with us if you have further questions.
> > >
> > > Best,
> > >
> > > Paper 383 Authors

---

### Official Review · Reviewer_itxM · 2023-11-09

**Soundness:** 3 good
**Presentation:** 3 good
**Contribution:** 2 fair
**Rating:** 5
**Confidence:** 4

**Summary:**

This paper builds upon the foundations laid by ProgPrompt by introducing three key mechanisms: (1) prompt ensemble, (2) prompt dequeuing, and (3) shared prompt combined with memory retention. The novelty appears somewhat limited as it combines elements from several existing works. Further empirical results are necessary to thoroughly assess the effectiveness of the proposed method and identify its true impact.

**Strengths:**

- The research topic of continual learning for pre-trained models is of utmost importance.
- The existing empirical results demonstrate that the proposed method attains a new state-of-the-art performance level.
- The paper is well-written and easy to follow.

**Weaknesses:**

- The novelty of the proposed method appears to be relatively limited, as the newly introduced key modules have all been explored by other works. Surprisingly, the authors do not engage in discussions or comparisons with those existing works.
  * In [1], although adapters are used, a prompt essentially represents a specialized version of an adapter. The authors in that work have already addressed crucial questions such as (1) how to select a suitable adapter, and (2) how to integrate a newly adapted version customized for the current task into the pool of adapters, a concept akin to the prompt queue in this study. It's worth noting that the argument made by the authors in Line 150 regarding "the quantitative correlation of different tasks is hard to define" has been resolved in this work.
  * The effectiveness of the shared prompt concept has been demonstrated in the DualPrompt work [2].
  * The ensemble approach involving existing prompts, as investigated and proven effective in CODA-Prompt [3], has not been acknowledged or discussed in this context.
- Some important empirical results to justify the proposed method are missing.
  *Exploration using more advanced language models, such as LLaMA, could determine whether the algorithm is sensitive to different backbones.
  * Ablation studies focusing on the prompt ensemble, utilizing only a linear combination of prompts instead of incorporating a weighting matrix with numerous parameters, would offer valuable insights.
  * Given the introduction of additional parameters, a fair comparison with ProgPrompt could be conducted by increasing its prompt length.
  * To provide a comprehensive analysis, it would be beneficial to extend the comparison by adding a shared prompt to ProgPrompt and evaluating the results in this context.


[1] Ermis, Beyza, et al. "Memory efficient continual learning with transformers." Advances in Neural Information Processing Systems 35 (2022): 10629-10642.
[2] Wang, Zifeng, et al. "Dualprompt: Complementary prompting for rehearsal-free continual learning." European Conference on Computer Vision. Cham: Springer Nature Switzerland, 2022.
[3] Smith, James Seale, et al. "CODA-Prompt: COntinual Decomposed Attention-based Prompting for Rehearsal-Free Continual Learning." Proceedings of the IEEE/CVF Conference on Computer Vision and Pattern Recognition. 2023.

**Questions:**

- Could you kindly explain why the first term of equation (5) does not include Q^i together with the shared prompt? Have you experimented with this version, and if the performance was unsatisfactory, could you provide insights into the reasons behind it?
- It appears that enlarging the queuing size results in marginal performance improvement. Could you elaborate on the reasons for this phenomenon?
- MTL is expected to serve as an upper bound, but it is puzzling that its performance is even lower than that of the per-task prompt. Could you offer insights into this unexpected outcome or explain your detailed implementation for it?

---

> ### Author Response · Authors · 2023-11-15
> **To Reviewer itxM:**
>
> Thank you very much for your detailed and constructive review of our paper and all the references you have provided! We are encouraged to know that you found our idea promising and our paper well-written and well-validated. We have revised our paper to address your concerns, and we also added the papers you recommended in our related work section. In order to clarify our core contributions and resolve all your questions, please see our answers below:
>
> ---
>
> **Q1**: **The novelty of the proposed method appears to be relatively limited, as the newly introduced key modules have all been explored by other works.**
>
> **A1**: Thank you for introducing these relevant papers, and we've added them to the related work (Lines 77-78 and 87-88). We'd like to elaborate on the difference between Q-Tuning and the mentioned papers:
>
> 1. The paper [1] requires buffer data from previous tasks for the distillation training; hence, it's not rehearsal-free. Q-tuning focuses on the scenario where we no longer have access to historical data;
>
> 2. A major contribution of DualPrompt [2] is to add the shared prompt and task-specific prompts to different layers of the backbone model. This layer-wise prompt configuration, however, necessitates the **prior determination of the total number of tasks**, which is orthogonal to our approach, as Q-Tuning's focus is on enabling lifelong learning where **the total number of tasks is unknown and could be very large**.
>
> While acknowledging a conceptual similarity with their framework (shared prompt + task-specific prompts), Q-tuning distinguishes itself by excelling in lifelong learning tasks. This success can be attributed to the elaborate memory retention loss for the shared prompt and the application of the Q-prompt with the De-Q rule. Indeed, we have compared Q-tuning with a configuration similar to the DualPrompt which uses a shared prompt and a task-specific prompt for each task. The results are reported in Tables 5 & 6 (extremely long sequence of experiments, including **70 different tasks**) in the paper. We copy them here for reference.
>
> | Method | Order11 | Order12 | Order13 | Average |
> | -------- | -------- | -------- | -------- |-------- |
> | Prompt list (w/o Q-prompt) | Out of Memory | Out of Memory | Out of Memory | -
> | Q-prompt (appending task-specific promtps)     | 86.8 | 87.3 | 87.7 |  87.3
>  Q-prompt + shared prompt  | 87.6 | 88.1 | 88.7 | 88.1 |
>  Q-prompt + shared prompt +Ensemble   | 89.8 | 89.4 | 90.1 | 89.8 |
>  Q-prompt + shared prompt +Ensemble + MR    | **90.9**| **90.6**| **90.8** | **90.8**
>
> The experiments are conducted on an NVIDIA V100 GPU 32GB.
> Observing the results in the table reveals that training fails unless the proposed Q-prompt with De-Q is adopted.
> By introducing Ensemble + MR to Q-prompt, we observe a performance increase from 88.1 to 90.8
>
> 3. We agree that the ensemble strategy itself is not a novel concept, as evidenced by its widespread use in machine learning (e.g., [3]). We emphasise that the core innovation of our approach lies in designing Q-tuning to facilitate lifelong prompt tuning for an unknown number of incoming tasks. Our current prompt ensemble approach represents one of several reasonable designs for adaptively incorporating previous prompts based on their relevance to the current task.
>
> ---
>
> **Q2**: **Ablation studies focusing on the prompt ensemble, utilizing only a linear combination of prompts instead of incorporating a weighting matrix with numerous parameters, would offer valuable insights.**
>
> **A2**: The weight matrix ${\mathcal{W}}^i$ is of **rank-one** (${\mathcal{W}}^i = \bf{u}_i \otimes \bf{v}_i^{\mathrm{T}}$, where $\bf{u}_i \in \mathbb{R}^{c^i}$, $\bf{v}_i \in \mathbb{R}^{d}$ and $\otimes$ denotes the **outer product**.) and hence the number of trainable parameters of ${\mathcal{W}}^i$ is $c^i+d$, where $c^i = l\times i$ is the total prompt length of the Q-prompt ${\mathcal{Q}}^i$ and $d$ is the feature dimension of the each prompt. Therefore, the parameter number of the weighting matrix ${\mathcal{W}}^i$ is very small in our experiments, e.g. $c^i+d = 10+1024$ for the short sequence experiments. In this context, **the Q-prompt ensemble using the rank-one weight matrix ${\mathcal{W}}^i$ is a linear combination of the prompts** in terms of the parameter complexity. We acknowledge that there could be different ensemble methods. We do not cover all of them, as they are not the focus of our work.

---

> ### Author Response · Authors · 2023-11-15
> **(Continued) To Reviewer itxM:**
>
> **Q3**: **A fair comparison with ProgPrompt could be conducted by increasing its prompt length.**
>
> **A3**: We actually made a fair comparison.  The total number of parameters in Q-tuning is actually fewer than ProgPrompt. As mentioned in Appendix C.3 - Table 10, both the **length of prompt per task and the length of the shared prompt are set to 10**. In comparison, in ProgPrompt, **the length of prompt per task is set to 50 (by copying the configuration of ProgPrompt to run their experiments)**. Therefore Q-tuning not only achieves better performance but also requires fewer trainable parameters.
>
> The weight matrix ${\mathcal{W}}^i$ is of rank-one (${\mathcal{W}}^i = \bf{u}_i \otimes \bf{v}_i^{\mathrm{T}}$, where $\bf{u}_i \in \mathbb{R}^{c^i}$, $\bf{v}_i \in \mathbb{R}^{d}$ and $\otimes$ denotes the **outer product**.) and hence the number of trainable parameters of ${\mathcal{W}}^i$ is $c^i+d$, where $c^i = l\times i$ is the total prompt length of the Q-prompt ${\mathcal{Q}}^i$ and $d$ is the feature dimension of the each prompt. The prompts in the Q-tuning and the baseline ProgPrompt are **both parameterized by a  2-hidden-layer MLP**. Hence, we can easily know that the number of parameters of ${\mathcal{W}}^i$ i.e. $c^i+d$ << the number of parameters of a  2-hidden-layer MLP.
>
>
> Here, we provide an example to compare the number of parameters for inference. Let us consider the experiments for short sequences (in Table 1 (a)) which include 4 different tasks. In these experiments, Q-tuning sets each prompt length $l=10$ while the ProgPrompt sets $l=50$. Both of them set the feature dimension $d=1024$. For inference, they both use the MLP's output as the prompts. In this context, the total  parameter numbers are listed as follows:
>
> | Method  | ${\mathcal{W}}^i$ | Shared Prompt| Prompt Buffer| Total Prompt | Average Accuracy
> | -------- | -------- | -------- | -------- |-------- |-------- |
> | ProgPrompt  | 0    |  0   | 50 x 1024 x4   |  **204,800** | **74.5**
> | Q-tuning  | 40+1024  | 10x1024   | 10x1024 x 4| **52,264** | **76.2**
>
> We can observe that the total number of parameters of ProgPrompt is **204,800** > **52,264** of Q-tuning while the performance of ProgPrompt **74.5** < **76.2** of Q-tuning.
>
> Furthermore, in general, we agree with the reviewer that the performance of prompt tuning significantly depends on the number of trainable parameters. However, the experiments in our paper are based on the **few-shot** continual learning settings, such as 20 samples/class. In this context, we empirically find that the performance is not sensitive to the prompt length. We think this is owing to the **MLP-based prompts** and the **few-shot setting**. Even though the input prompt is of a small length, the learning ability of MLP-based prompts is still powerful enough to achieve few-shot learning.
>
> The following table shows the T5 experiments by setting different lengths for each prompt in Q-tuning.
>  | Length | Average Accuracy |
>  | ------------- | ---------------- |
>  |    5          |       76.1      |
>  |    10         |       76.2      |
>  |    30         |       76.3      |
>  |    50         |       76.1      |
>
> In this exploration of various settings, we can identify marginal differences. For the sake of maintaining consistency across our short, long, and extremely long sequence experiments, we set the prompt length to 10 in our Q-tuning experiments.
>
> ---
>
> **Q4**: **To provide a comprehensive analysis,... by adding a shared prompt to ProgPrompt and evaluating the results in this context.**
>
> **A4**: We have conducted such a comparison in Table 5 (top rows) for the short sequence tasks. In these short sequence experiments, Q-tuning by dropping ensemble and shared prompt is the same as the ProgPromt.
>
> | Method | 16 samples | 200 samples | 1000 samples| Average
> | -------- | -------- | -------- | -------- | -------- |
> | ProgPrompt     | 74.5 | 79.8 | 79.8| 78.0 |
> | + Ensemble   | 75.2 | 80.9 | 80.4 | 78.8 (+0.8)|
> | + Shared Prompt   | 75.1| 80.6| 80.9| 78.9 (+0.9)|
> | + Ensemble & Shared Prompt   | 76.2 | 81.2| 81.9| 79.7 (+1.7)|
>
> We can observe a 1.7% improvement by adding Ensemble and Shared Prompt to the ProgPrompt. Furthermore, we want to highlight that the ProgPromt cannot achieve lifelong learning as demonstrated by our experiments on the extremely long sequences including Orders 11, 12 and 13 (Table 3).
>
> | Method | Order11 | Order12 | Order13 | Average |
> | -------- | -------- | -------- | -------- |-------- |
> | ProgPrompt | Fail | Fail | Fail | -
> | Per-task Prompt | 60.4 |60.4 |60.4| 60.4|
> | Shared Prompt | 62.4  | 62.7|  63.1| 62.7|
> |Q-tuning ($Q_{size} = 10$) | 90.9 |  90.6|90.8 | 90.8|
>
> We observe that training ProgPrompt will fail (after the 15th task on a single V100 GPU (32GB)) due to out of memory caused by the accumulation of prompts. Compared to the per-task prompt tuning, Q-tuning has gained considerable performance benefits (30.4\% accuracy improvement on average from 60.4\% to 90.8\%).

---

> ### Author Response · Authors · 2023-11-15
> **(Continued) To Reviewer itxM:**
>
> **Q5**: **Could determine whether the algorithm is sensitive to different backbones**.
>
> **A5**: All of our experiments are conducted by following the benchmark empirical settings for fair comparison.  The proposed Q-tuning is a **model-agnostic** method, as verified by the experiments using the T5 and Bert backbones (Table 1). If using the backbone of LLaMA, the current prompt-tuning approaches will also fail in life-learning scenarios, while our Q-tuning can work well. However, due to
> limited hardware resources, we cannot finish the **continual** prompt tuning experiments using the 65B-parameter model. In the continual learning setting, four or more prompts should be simultaneously maintained, which are too expensive to afford by our computation resource. Since popular benchmarks in the NLP domain are mainly evaluated on T5 models, we follow the same practice.  We'll evaluate Q-tuning in other large models (e.g., LLaMA) in our future work after getting access to instances with larger GPU memory.
>
> ---
>
> **Q6**: **Why the first term of equation (5) does not include $\mathcal{Q}^i$ together with the shared prompt?**
>
> **A6**: The shared prompt $\theta^i_{\mathcal{P}^{\ast}}$ is designed to carry the global knowledge across all previous tasks, so that we use equation (5) to avoid information loss after trimming the Q-prompt $\mathcal{Q}^i$ by the De-Q. For the $i$-th task, the Q-prompt $\mathcal{Q}^i = [\theta^1_{\mathcal{P}},\cdots, \theta^i_{\mathcal{P}}]$, where **only the new prompt $\theta^i_{\mathcal{P}}$ is trainable while the rest of prompts $[\theta^1_{\mathcal{P}},\cdots, \theta^{i-1}_{\mathcal{P}}]$ are fixed**. In this context, the trainable
> $\theta^i_{\mathcal{P}}$ in the Q-prompt is appended to learn the new task information rather than memorizing the old information.  If we add the $\mathcal{Q}^i$ to the first term of Equation (5), the knowledge information from the new task learned by the trainable prompt $\theta^{i-1}_{\mathcal{P}}$ in $\mathcal{Q}^i$ will be regularized, dropping the performance on the new task.
>
>
> ---
>
> **Q7**: **Enlarging the queuing size results in marginal performance improvement**
>
> **A7**:  We do not enlarge the queuing size to improve performance. The queue $\mathcal{Q}^i$ is used to maintain the trained prompts. The size of an untrimmed queue should be the same as the number of currently observed tasks. If the queuing size exceeds the maximum capability, we will apply a De-Q rule to trim the queue by discarding less informative prompts in the queue.
>
> In Tables 2&3, we conducted experiments on the 15 sequential tasks to evaluate the performance of the queue with two smaller queue sizes.  We can observe that, either setting the size to 10 (throwing away 10 prompts) or  5 (throwing away 5 prompts), the performance drops marginally compared to the Full Q-prompt with $\mathcal{Q}_{size}=15$, because the proposed De-Q algorithm can retain the most information in the queue. **The marginal performance differences between these settings are expected, which demonstrate the effectiveness of the novel Q-prompt and the De-Q algorithm.**
>
> ---
>
> **Q8**: **MTL is expected to serve as an upper bound, but it is puzzling that its performance is even lower than that of the per-task prompt.**
>
> **A8**: MTL is not necessarily an upper bound. Indeed, the performance of MTL varies across different types of language models. BERT adopts an encoder-only structure, which requires training a nonlinear classifier head for each task. T5 is a text-to-text generative model that does not need a nonlinear head for each task. Hence, when conducting the MTL training, for example, for 4 different tasks, the BERT model should have  4 different classifier heads that are optimized respectively. However, the T5 model is trained to learn the average information of the 4 different tasks, to some extent. Therefore, we observe that the MTL's performance is the best in BERT-based experiments but not in T5-based ones. Besides, MTL's performance also depends on the relevance of tasks. Tasks vary a lot in the NLP domain. E.g., our experiments cover topics like yp reviews, banking, emotion, etc. The negative correlation among tasks might hurt the performance of MTL. In comparison, Q-Tuning leverages the Q-prompt Ensemble which adaptively incorporates previous prompts to avoid this issue.
>
> ---
>
> [1] Ermis, Beyza, et al. "Memory efficient continual learning with transformers." Advances in Neural Information Processing Systems 35 (2022): 10629-10642.
>
> [2] Wang, Zifeng, et al. "Dualprompt:Complementary prompting for rehearsal-free continual learning." European Conference on Computer Vision. Cham: Springer Nature Switzerland, 2022.
>
> [3] Smith, James Seale, et al. "CODA-Prompt:COntinual Decomposed Attention-based Prompting for Rehearsal-Free Continual Learning."

---

> > ### Author Response · Authors · 2023-11-19
> > **Followup Response to Reviewer itxM**
> >
> > Dear Reviewer itxM:
> >
> > We express our sincere gratitude again for your invaluable efforts in reviewing our paper. In our response, we have clarified the novelty of our paper and comprehensively analyzed the difference between our paper and the previous techniques that you mentioned. We also provided a comprehensive discussion of the empirical findings you highlighted. Please do not hesitate to let us know if there are additional questions; we would be more than happy to help with them.
> >
> > Best,
> >
> > Paper 383 Authors

---

> > ### Comment · Reviewer_itxM · 2023-11-22
> > **Thank the authors' detailed response**
> >
> > I appreciate the authors' thorough response, yet I still have concerns regarding (1) the core technical contributions, which appear to be a combination of existing ideas in the literature, and (2) the veracity of the proposed method's efficacy as claimed, especially considering the responses to Q3/Q7. Consequently, I maintain my current score.

---

> ### Author Response · Authors · 2023-11-22
> **Response to Reviewer itxM**
>
> Thank you for your response. As clarified in our earlier rebuttal Q1&A1, the **core contributions of the paper are the Q-prompt, MR loss, and the DQ-PCA algorithm, which are not a simple combination of exiting techniques**. Our extensive experiments in Tables 2&3 demonstrate the effectiveness of the proposed techniques. We believed that Q3&Q7 had been addressed in our earlier rebuttal. The following response may help us to clarify further.
>
> 1. Although we introduce additional parameters in Q-prompt Ensemble and Shared Prompt, our total number of parameters is still less than the baseline (i.e., ProgPrompt). This is because **ProgPrompt set the prompt length as 20 for BERT and 50 for T5, whereas we only set it to 10**  (check Appendix Table 10 of our paper and the Appendix Table 7 of the ProgPrompt paper). Our experiments show that our Q-tuning outperforms the baseline (74.5 vs. 76.2), therefore we believe our better performance than ProgPrompt is not simply because of adding more parameters. In our previous rebuttal (Q3 & A3), we conducted a detailed analysis of the number of parameters and provided an explanation for the effects of setting different prompt lengths in our experiments.
>
> Moreover, we want to call out that our core contribution that set us apart from existing approaches is the capability of real lifelong learning (i.e., extremely long task sequence such as 70) through DQ-PCA, while avoiding losing previous information through shared prompt and MR. E.g., for the 70-task sequence, ProgPrompt (our baseline) will fail due to the lack of a mechanism to clean up cumulated prompts, as the following table reports:
>
> Experiments on 70 tasks:
>
> | Method | Order11 | Order12 | Order13 | Average |
> | -------- | -------- | -------- | -------- |-------- |
> | ProgPrompt | Fail | Fail | Fail | -
> |Q-tuning ($Q_{size} = 10$) | 90.9 |  90.6 | 90.8 | 90.8|
>
> 2. Marginal performance improvement by enlarging the $Q_{size}$.
>
> | Method | Order8 | Order9 | Order10 | Average |
> | -------- | -------- | -------- | -------- |-------- |
> | Q-tuning ($Q_{size} = 5$ + Random) |76.4| 77.3 |76.1 | 76.6 (-2.1)|
> | Q-tuning ($Q_{size} = 5$ + DQ-PCA) |77.5| 78.8 |77.8 |78.0 (-0.7)|
> |Q-tuning ($Q_{size} = 10$ + Random) | 76.7 | 77.2| 76.5 |76.8 (-1.9)|
> |Q-tuning ($Q_{size} = 10$ + DQ-PCA) | 78.3 | 79.7 | 78.7 | 78.9 (+0.2)|
> |Q-tuning ($Q_{size} = 15$ (Full)) | 79.0 | 79.1 | 78.1 | 78.7 |
>
> This is expected as DQ-PCA ensures that only the least informative previous prompts are thrown away. Table 2 (copied here) demonstrates the efficacy of DQ-PCA compared with other naive approaches of throwing away previous prompts. Compared to the full Q-prompt, randomly evicting prompts in queue causes -2.1\% and -1.9\% performance drops for $Q_{size} = 5$ and $Q_{size} = 10$, respectively. However, when using the proposed DQ-PCA, the performance of the trimmed Q-prompt matches that of the full Q-prompt, showcasing the effectiveness of the DQ-PCA algorithm in preventing information loss when trimming the Q-prompt (see experiments when setting  $Q_{size}=5$). Moreover, since DQ-PCA can **remove some redundant information in previously trained prompts**, it also serves a data cleaning role on previous prompt, which can improve the performance to some extent. Therefore, the marginal performance improvement resulting from enlarging the $Q_{size}$ of Q-tuning aligns with our expectations.
>
> We hope that we have addressed your concerns, and we would greatly appreciate it if you could adjust your rating of our paper.

---

### Official Review · Reviewer_KTxv · 2023-11-09

**Soundness:** 3 good
**Presentation:** 3 good
**Contribution:** 3 good
**Rating:** 5
**Confidence:** 4

**Summary:**

This paper introduces a new approach for continual learning, called Q-tuning, based on prompt tuning. Specifically, the authors introduce a dynamic prompt ensemble mechanism to adaptively incorporate the knowledge within previous prompts, according to a given task. In addition, to address the computational limitation from the progressive addition of the prompts, they introduce two different mechanisms, the De-Q Rule and globally shared prefix; the De-Q Rule selectively removes the most irrelevant parts from the currently constructed prompts. The globally shared prefix is distilled from the previous set of prompts with memory retention regularization to compensate for the information loss during De-Q. With two popular language models (T5 and BERT), Q-tuning has been demonstrated in the various experimental scenarios, consistently improving the strong baselines in the field.

**Strengths:**

1. **Clarity**. Overall, the writing is clear and easy to follow. In addition, the organization of the main draft is well-established.
2. **Well motivated problem**. Improving the continual prompt tuning with LMs is an interesting and important problem. To this end, overcoming the previous limitation from the increase of tasks is a reasonable and well-motivated direction.
3. **Intuitive method.** The proposed method is intuitive and seems to be applicable to various LMs. Also, it shows consistent improvement compared to the existing baselines, and the gain is significantly enlarged with challenging setups.

**Weaknesses:**

1. **Fairness with baselines**. The proposed Q-tuning uses more trainable parameters by introducing $W^i$ for dynamic prompt ensemble mechanism and $P*$ for global knowledge sharing. Since the performance of prompt tuning significantly depends on the number of trainable parameters, the author should compare the baselines that use the same number of trainable parameters for a fair comparison. For example, increasing the prompt length for the baselines, to match the number of parameters. To be specific, it is doubtable that the gain of Q-tuning over ProgPrompt in Table 1 is the consequence of this, as there is no issue regarding the information loss and De-queuing here.
2. **Consistency of presentation.** The experimental results are presented inconsistently across the tables. For example, Tables 1 and 2 report the average over 3 runs. But, Tables 3 and 6 suddenly report the variance without mentioning the reason and the number of runs. Also, for the tables that omitted to report the variance, there are no results with variance in the Appendix. To provide the information to readers, it seems to be included in the main tables or the appendix with proper description.

### Minor

1. **Typos.**  Lines 132 and 137: $W$ → $W^i$.
2. **Editorial comments.** In Figure 1, there is no mention of $W^i$ in the caption while it is presented in the figure. To enhance the readability, it seems to be included.

**Questions:**

Please address the concerns in Waeknesses, especially on **Fairness with baselines**. If the concerns are properly addressed, I'm willing to increase the score.

---

> ### Author Response · Authors · 2023-11-15
> **To Reviewer KTxv**
>
> Thank you very much for your constructive review and feedback! We are encouraged to learn that you found our proposed method well-motivated and intuitive. Here we provide the answers to your questions and comments as follows:
>
> ---
> **Q1:** **Fairness with baselines. The proposed Q-tuning uses more trainable parameters by introducing $W^i$ for dynamic prompt ensemble mechanism and $P^{\ast}$ ... depends on the number of trainable parameters, the author should compare the baselines that use the same number of trainable parameters for a fair comparison**
>
> **A1**: The total number of parameters in Q-tuning is actually **fewer** than ProgPrompt. As mentioned in Appendix C.3 - Table 10, both the **length of the prompt per task and the length of the shared prompt are set to 10**. In comparison, in ProgPrompt, **the length of prompt per task is set to 50 (by copying the configuration of ProgPrompt to run their experiments)**. Therefore, Q-tuning not only achieves better performance but also requires fewer trainable parameters.
>
> The weight matrix ${\mathcal{W}}^i$ is of **rank-one** (${\mathcal{W}}^i = \bf{u}_i \otimes \bf{v}_i^{\mathrm{T}}$, where $\bf{u}_i \in \mathbb{R}^{c^i}$, $\bf{v}_i \in \mathbb{R}^{d}$ and $\otimes$ denotes the **outer product**.) and hence the number of trainable parameters of ${\mathcal{W}}^i$ is $c^i+d$, where $c^i = l\times i$ is the total prompt length of the Q-prompt ${\mathcal{Q}}^i$ and $d$ is the feature dimension of the each prompt. The prompts in the Q-tuning and the baseline ProgPrompt are **both parameterized by a 2-hidden-layer MLP**. Hence, we can easily know that the number of parameters of ${\mathcal{W}}^i$ i.e. $c^i+d$ << the number of parameters of a 2-hidden-layer MLP for training.
>
> Here, we provide an example to compare the number of parameters for inference. Let us consider the experiments for short sequences (in Table 1 (a)) which include 4 different tasks. In these experiments, Q-tuning sets each prompt length $l=10$ while the ProgPrompt sets $l=50$. Both of them set the feature dimension $d=1024$. For inference, they both use the MLP's output as the prompts. In this context, the total numbers of the parameters are listed as follows:
>
> | Method  | ${\mathcal{W}}^i$ | Shared Prompt| Prompt Buffer| Total Prompt | Average Accuracy
> | -------- | -------- | -------- | -------- |-------- |-------- |
> | ProgPrompt  | 0    |  0   | 50 x 1024 x4   |  50 x 1024 x4 = **204,800** | **74.5**
> | Q-tuning  | 40+1024  | 10x1024   | 10x1024 x 4| 10x1024 x 4 + 10x1024 + 40+1024 = **52,264** | **76.2**
>
> We can observe that the total number of parameters of ProgPrompt is **204,800** > **52,264** of Q-tuning, while the performance of ProgPrompt is **74.5** < **76.2** of Q-tuning.
>
> Furthermore, in general, we agree with the reviewer that the performance of prompt tuning significantly depends on the number of trainable parameters. However, the experiments in our paper are based on the **few-shot** continual learning settings, such as 20 samples/class. In this context, we empirically find that the performance is not sensitive to the prompt length. We think this is owing to the **MLP-based prompts** and the **few-shot setting**. Even though the input prompt is of a small length, the learning ability of MLP-based prompts is still powerful enough to achieve few-shot learning.
>
> The following table shows the T5 based Q-tuning experiments by setting different lengths for each prompt.
>
>  | Length | Average Accuracy |
>  | ------------- | ---------------- |
>  |    5          |       76.1      |
>  |    10         |       76.2      |
>  |    30         |       76.3      |
>  |    50         |       76.1      |
>
> In this exploration of various settings, we can identify marginal differences. For the sake of maintaining consistency across our short, long, and extremely long sequence experiments, we set the prompt length to 10 in our Q-tuning experiments.
>
> ---
> **Q2**: **The gain of Q-tuning over ProgPrompt in Table 1**
>
> **A2**: We believe the gain of Q-Tuning mainly comes from the following two aspects:
> 1. For a new given task, the Q-prompt Ensemble module incorporates previous prompts adaptively based on their relevance to the task;
> 2. The shared prompt compliments the task-specific prompts in the Q-prompt. For a given new task, the task-specific prompt is responsible for carrying the incremental information that is not captured by previous prompts, whereas the shared prompt carries the global information across all the tasks.
> Here, we show the ablation study results on orders 1,2, and 3 as reported in Table 5 in the paper.
>
> | Method | 16 sampels/class| 200 sampels/class | 1000 sampels/class| Average
> | -------- | -------- | -------- |-------- |-------- |
> | Only Q-prompt      |  74.5 | 79.8 |79.8 | 78.0
> | Q-prompt + $\theta_{\mathcal{P}^{\ast}}$     |  75.1 | 80.6|80.9 | 78.9 (+0.9)
> | Q-prompt + ensemble  |75.2| 80.9| 80.4| 78.8 (+0.8)
>
> More results of each order are reported in Appendix Table 11.

---

> > ### Author Response · Authors · 2023-11-15
> > **(Continued) To Reviewer KTxv**
> >
> > **Q3**: **Consistency of presentation. For example, Tables 1 and 2 report the average over 3 runs. But, Tables 3 and 6 suddenly report the variance ... omitted to report the variance...**
> >
> > **A3**: Thank you for your suggestion. First, we want to clarify that all the experiments in the paper are based on the results of 3 runs. For Tables 1 & 2, we followed the same experimental settings as the ProgPrompt paper for all the baselines, so that we only report the average result without the variance following the same format.
> >
> > The reason why we reported variances in Tables 3 & 6 is that we initially thought that variance might be more significant in extremely long sequence experiments. But now, based on the reported values, the variances for extremely long sequence experiments are comparably small. We have removed them from Tables 3 & 6 in the revised paper for consistency with Tables 1 & 2.
> >
> > ---
> >
> > **Q4**: **Typos and unclear captions**
> >
> > **A4**: We've revised the typos and unclear captions in the updated paper.
> >
> > ---
> > Finally, we want to highlight the results reported in Table 3, which demonstrate the effectiveness of the proposed Q-tuning on extremely long tasks. The previous SOTA baseline method fails in this scenario. This provides strong proof to support the claimed core technical contribution in the paper, i.e., enabling life-long prompt tuning with LMs.

---

> > > ### Author Response · Authors · 2023-11-19
> > > **Followup Response to Reviewer KTxv**
> > >
> > > Dear Reviewer KTxv:
> > >
> > > Thanks again for your valuable suggestions, and we sincerely appreciate your acknowledgement of our work. We hope our clarifications on the fairness of baselines can solve your concerns. Please feel free to share with us if you have any more questions.
> > >
> > > Best,
> > >
> > > Paper 383 Authors